# Uncertainty Herding: One Active Learning Method for All Label Budgets

**Wonho Bae**
University of British Columbia & Borealis AI
whbae@cs.ubc.ca

**Danica J. Sutherland**\*
University of British Columbia & Amii
dsuth@cs.ubc.ca

**Gabriel L. Oliveira**\*
Borealis AI
gabriel.oliveira@borealisai.com

## Abstract

Most active learning research has focused on methods which perform well when many labels are available, but can be dramatically worse than random selection when label budgets are small. Other methods have focused on the low-budget regime, but do poorly as label budgets increase. As the line between "low" and "high" budgets varies by problem, this is a serious issue in practice. We propose *uncertainty coverage*, an objective which generalizes a variety of low- and high-budget objectives, as well as natural, hyperparameter-light methods to smoothly interpolate between low- and high-budget regimes. We call greedy optimization of the estimate Uncertainty Herding; this simple method is computationally fast, and we prove that it nearly optimizes the distribution-level coverage. In experimental validation across a variety of active learning tasks, our proposal matches or beats state-of-the-art performance in essentially all cases; it is the only method of which we are aware that reliably works well in both low- and high-budget settings.

## 1 Introduction

In active learning, rather than being provided a dataset of input-output pairs as in passive learning, a model strategically requests annotations for specific unlabeled inputs. The aim is to learn a good model while minimizing the number of required output annotations. This procedure is generally iterative: a model is initially trained on a small, labeled dataset, then selects the most "informative" data points from an unlabeled pool to annotate. This is particularly useful when labeling is expensive or time-consuming. For example, manual annotations of medical imaging by radiologists or pathologists may be especially time-consuming and costly. Measuring whether a compound interacts with a certain biological compound may require slow, high-accuracy chemical simulations or even lab experiments. Discovering a customer's product preferences may require giving them many offers, which is slow, potentially expensive, and may produce a poor customer experience.

The most popular line of work in active learning has used notions of uncertainty to measure how informative each candidate data point is expected to be, and selects data points for labeling to maximize that measure. Although these uncertainty-based approaches often work well in the experimental settings where they are evaluated, Hacohen et al. (2022); Yehuda et al. (2022) have shown that in low-budget regimes–where budget refers to the total number of labeled data points–they can be substantially worse than random selection, presumably because the model's estimate of uncertainty is not yet reliable. To address this, they (and some others) have proposed methods that prioritize "representative" data points, often built on clustering methods such as $k$-means. These methods can work substantially better in low-budget regimes, but themselves often saturate performance and do worse than uncertainty-based selection once budgets are large enough.

In practice it is difficult to know whether a given budget is "high" or "low" for a particular problem; it greatly depends on the particular dataset and model architecture. Hacohen & Weinshall (2024)

---

\*These authors contributed equally.

proposed an algorithm, SelectAL, to select whether to use a high- or low-budget method. This approach, however, assumes discrete budget regimes when there is often not a clear boundary, and because of the form of the algorithm is also unable to consider uncertainty-based active learning measures directly. SelectAL also requires re-training models many times, which may be computationally infeasible, and requires a nontrivial amount of data holdout for validation, an issue when budgets are low. Perhaps most importantly, the algorithm appears quite sensitive to small, subtle decisions; our attempts at replication[1] gave extremely inconsistent and unreliable estimates of the regime, overall yielding worse performance than reported in the paper.

This motivates the aim of finding a single active learning algorithm which can seamlessly adapt from low- to high-budget regimes. While there have been various hybrid methods combining representation and uncertainty, we find in practice that none of these methods work well in low-budget settings. We therefore propose an objective called *uncertainty coverage*, adding a notion of uncertainty to the "generalized coverage" of Bae et al. (2024) and its greedy optimizer MaxHerding. We call greedy optimization of the empirical estimate of the uncertainty coverage, which we prove nearly optimizes the true uncertainty coverage, Uncertainty Herding (UHerding).

The uncertainty coverage agrees with the generalized coverage in one extreme of parameter setting, while agreeing with uncertainty measures in another. To naturally interpolate between those settings, we propose a simple method to adaptively and automatically adjust these parameters such that the objective moves itself from mostly representation-based to mostly uncertainty-based behavior. With this parameter adaptation scheme, we demonstrate that UHerding outperforms MaxHerding (and all other methods) in low-budget regimes, while also outperforming uncertainty sampling (and all other methods) in high-budget regimes, across several benchmark datasets (CIFAR-10 and -100, Tiny-ImageNet, DomainNet, and ImageNet) in both standard supervised and transfer learning settings. Furthermore, we describe how several existing hybrid active learning methods are closely related to UHerding, and confirm that our parameter adaptation schemes also benefit existing hybrid methods.

## 2 RELATED WORK AND BACKGROUND

The most common framework in active learning is pool-based active learning: at each step $t \in \{1, 2, \cdots, T\}$, a labeled set $\mathcal{L}_t \subseteq \mathcal{X}$ is iteratively expanded by querying a set of new data points $\mathcal{S}_t = \{\mathbf{x}_b\}_{b=1}^{B_t}$ from an unlabeled pool of data points $\mathcal{U}_t \subseteq \mathcal{X}$ where $\mathcal{X}$ denotes the support set of the data distribution of interest. A model is then trained on the new $\mathcal{L}_{t+1}$. Usually, the most important component is determining which points to annotate.

**Uncertainty-based Methods** "Myopic" methods that rely only on a current model's predictions include entropy (Wang & Shang, 2014), margin Scheffer et al. (2001), confidence, and posterior probability (Lewis & Catlett, 1994; Lewis & Gale, 1994). In the Bayesian setting, BALD (Gal et al., 2017; Kirsch et al., 2019) uses mutual information between labels and model parameters. BAIT (Ash et al., 2021) tries to select data points that minimize Bayes risk. Instead of using a snapshot of a trained model, Kye et al. (2023) exploit uncertainties computed in the process of model training.

Some models focus on "looking ahead" to predict how a data point will change the model. These include methods based on expected changes in model parameters (Settles, 2009; Settles et al., 2007; Ash et al., 2020), expected changes in model predictions (Freytag et al., 2014; Käding et al., 2016; 2018), and expected error reduction (Roy & McCallum, 2001; Zhu et al., 2003; Guo & Greiner, 2007). These approaches are primarily used with simple models like linear and Naïve Bayes models. For deep models, a neural tangent kernel-based linearization (Mohamadi et al., 2022) can be used, though it offers limited improvement over uncertainty sampling relative to its computational cost.

**Representation-based Methods** These methods select data points that represent (or cover) the data distribution. Traditional methods include $k$-means (Xu et al., 2003), medoids (Aghaee et al., 2016) and medians (Voevodski et al., 2012). Hacohen et al. (2022) show that selecting representative points is particularly helpful in low-budget regimes, and propose Typiclust, which selects the "most typical" points in each cluster. Bıyık et al. (2019) utilize determinantal point processes instead.

---

[1] The authors have not publicly released code though indicated in private communication they plan to do so.

Another approach is to minimize distance between the labeled and unlabeled data distributions, whether kernel MMD (Chen et al., 2010), Wasserstein distance (Mahmood et al., 2022), or an estimate of the KL divergence tailored to transfer learning as in ActiveFT (Xie et al., 2023).

Sener & Savarese (2018) convert the objective of active learning into maximum coverage, proposing greedy $k$-center. Instead of finding the minimum radius to cover all data points, ProbCover (Yehuda et al., 2022) greedily select data points to cover the most data points with a fixed radius. While its performance is sensitive to the choice of radius (also difficult to set), MaxHerding (Bae et al., 2024) generalizes to a continuous notion of coverage, generalized coverage (or GCoverage), which is less sensitive to parameter choice. As we build directly on this method, we describe it in more detail.

GCoverage is defined in terms of a function $k : \mathcal{X} \times \mathcal{X} \times (\mathcal{X} \to \mathcal{V}) \to \mathbb{R}_{\geq 0}$, which computes a similarity between $\mathbf{x}$ and $\mathbf{x}'$ based on a feature mapping $g : \mathcal{X} \to \mathcal{V}$. Bae et al. (2024) mostly use the Gaussian kernel[2] $k_\sigma(\mathbf{x}, \mathbf{x}'; g) = \exp\left(-\|g(\mathbf{x}) - g(\mathbf{x}')\|^2 / \sigma^2\right)$ with $g$ based on self-supervised feature extractors such as SimCLR (Chen et al., 2020). The GCoverage and its estimator are

$$C_{k_\sigma}(\mathcal{S}) := \mathbb{E}_{\mathbf{x}}\left[\max_{\mathbf{x}' \in \mathcal{S}} k_\sigma(\mathbf{x}, \mathbf{x}'; g)\right] \approx \frac{1}{N} \sum_{n=1}^{N} \left(\max_{\mathbf{x}' \in \mathcal{S}} k_\sigma(\mathbf{x}_n, \mathbf{x}'; g)\right) =: \widehat{C}_{k_\sigma}(\mathcal{S}) \text{ with } \mathbf{x}_n \in \mathcal{U}. \quad (1)$$

MaxHerding greedily maximizes the estimated GCoverage: $\mathbf{x}^* \in \arg\max_{\tilde{\mathbf{x}} \in \mathcal{U}} \widehat{C}_{k_\sigma}(\mathcal{L} \cup \{\tilde{\mathbf{x}}\})$. This is a $\left(1 - \frac{1}{e}\right)$ approximation algorithm for optimizing the monotone submodular function $\widehat{C}_{k_\sigma}$.

**Hybrid Methods**    These methods aim to select informative yet representative data points. Nguyen & Smeulders (2004); Donmez et al. (2007) use margin-based selection weighted by clustering scores. Settles & Craven (2008) weight uncertainty measures like entropy by cosine similarity. For neural networks, BADGE (Ash et al., 2020) uses $k$-means++ on loss gradient space, while ALFA-Mix (Parvaneh et al., 2022) applies $k$-means to uncertain points based on feature interpolation.

Since uncertainty and representation-based active learning approaches behave differently in different budget regimes, SelectAL (Hacohen & Weinshall, 2024) and TCM (Doucet et al., 2024) provide methods to decide when to switch from low-budget to high-budget methods. TCM provides some insights for a transition point, but their insights are based on extensive experimentation, and hard to generalize to different settings. SelectAL was discussed in the previous section. In this work, we propose a more robust approach with minimal re-training, covering continuous budget regimes.

**Clustering in active learning**    $k$-means is widely used in active learning to promote diversity in selection. BADGE, ALFA-Mix, and Typiclust, for example, use $k$-means or variants.

$k$-means centroids, however, do not in general correspond to any available point, and thus other methods (such as Typiclust's density criterion) must be used to choose a point from a cluster. It would be natural to instead enforce centroids to be data points, yielding $k$-medoids. The common alternating update scheme similar to $k$-means often leads to poor local optima (Schubert & Rousseeuw, 2021). The Partitioning Around Medoids (PAM) algorithm (Kaufman & Rousseeuw, 2009; Schubert & Rousseeuw, 2019; Schubert & Lenssen, 2022) gives better clusters, but is much slower. MaxHerding selects points with essentially equivalent downstream performance as using the far more expensive Faster PAM algorithm (Schubert & Rousseeuw, 2021) for GCoverage.

As we demonstrate empirically in Figure 8b, MaxHerding achieves significantly better performance than both $k$-means and $k$-means++ for active learning. Thus, in Section 3.4, we replace $k$-means with MaxHerding for the theoretical analysis of some clustering-based active learning methods.

## 3 METHOD

We introduce a novel approach called Uncertainty Herding (UHerding), designed to "interpolate" between the state-of-the-art representation-based method MaxHerding and any choice of uncertainty-based method (*e.g.*, Margin), providing effectiveness across different budget regimes.

---

[2] Although we call the function $k_\sigma$ and use the term "kernel," it is not generally necessary that the function be positive semi-definite as in kernel methods, nor that it integrate to 1 as in kernel density estimation.

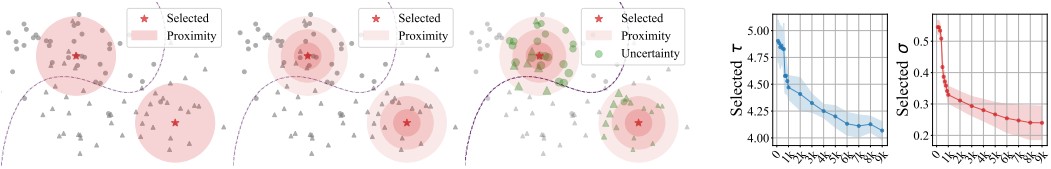

(a) Comparison of ProbCover, GCoverage, and UCoverage.

(b) Temperature $\tau$, lengthscale $\sigma$.

Figure 1: Left: illustration of coverages (Section 3.1). Right: parameter adaptation (Section 3.2).

## 3.1 UNCERTAINTY COVERAGE

We first define a measure of how much uncertainty a set of data points covers.

**Definition 1.** For any subset $\mathcal{S} \subset \mathcal{X}$, a nonnegative-valued function $k_\sigma$[3], and nonnegative-valued uncertainty function $U$, *uncertainty coverage* (UCoverage) is defined and empirically estimated as

$$\text{UC}_{k_\sigma}(\mathcal{S}) = \mathbb{E}_{\mathbf{x}}\left[U(\mathbf{x};f)\max_{\mathbf{x}'\in\mathcal{S}}k_\sigma(\mathbf{x},\mathbf{x}';g)\right] \approx \frac{1}{N}\sum_{n=1}^{N}U(\mathbf{x}_n;f)\max_{\mathbf{x}'\in\mathcal{S}}k_\sigma(\mathbf{x}_n,\mathbf{x}';g) = \widehat{\text{UC}}_{k_\sigma}(\mathcal{S}).$$

Here, the uncertainty function $U$ is based on a model $f$ which is updated as the model trains, while $k_\sigma$ uses a fixed feature extractor $g$. UCoverage weights the GCoverage of (1) with a choice of uncertainty measure $U(\mathbf{x};f)$; choosing $U(\mathbf{x};f) = 1$ immediately recovers GCoverage.

In Figure 1a, we visualize differences in assessing coverage. ProbCover (left) treats all data points within a $\sigma$-radius ball around selected points equally, assigning uniform weight to each. GCover (middle) introduces a smooth proximity measure by applying a kernel function, such as the RBF kernel, to weigh nearby points more effectively. UCover (right) additionally incorporates uncertainty. The green circles represent uncertainty, with their size proportional to each point's uncertainty. Ultimately, UCover evaluates how well the selected data points account for uncertainty across the space.

We now show $\widehat{\text{UC}}_{k_\sigma}$ is a good estimator of $\text{UC}_{k_\sigma}$. All proofs are in Appendix A.

**Theorem 2.** *Let* $U(\mathbf{x};f) \in [0, U_{\max}]$, $k_\sigma(\mathbf{x},\mathbf{x}';g) = \tilde{k}_\sigma(g(\mathbf{x}),g(\mathbf{x}')) \in [0,1]$, $\{g(\mathbf{x}) : \mathbf{x} \in \mathcal{U}\} \subseteq \{\mathbf{t} \in \mathbb{R}^d : \|\mathbf{t}\| \le R\}$, *and* $\left|\tilde{k}_\sigma(\mathbf{t},\mathbf{t}') - \tilde{k}_\sigma(\mathbf{t},\mathbf{t}'')\right| \le L_\sigma \|\mathbf{t}'-\mathbf{t}''\|$. *Let* $\mathcal{L} \subseteq \mathcal{X}$ *be arbitrary and fixed. Assume* $B/N < 16R^2$. *Then, with probability at least* $1-\delta$ *over the choice of the $N$ iid data points in* $\mathcal{U} \subseteq \mathcal{X}$ *used to estimate* $\widehat{\text{UC}}_{k_\sigma}$, *all size-$B$ sets $\mathcal{S}$ (not only subsets of $\mathcal{U}$) have low error:*

$$\sup_{\substack{\mathcal{S}\subseteq\mathcal{X}\\|\mathcal{S}|=B}}\left|\text{UC}_{k_\sigma}(\mathcal{L}\cup\mathcal{S}) - \widehat{\text{UC}}_{k_\sigma}(\mathcal{L}\cup\mathcal{S})\right| \le U_{\max}\sqrt{\frac{B}{N}}\left[8L_\sigma + \frac{1}{2}\sqrt{d\log\left(R^2\frac{N}{B}\right)} + \frac{2}{B}\log\frac{2}{\delta}\right].$$

Since typically $B \ll N$ – we query up to perhaps a few hundred points at a time, out of a dataset of at least tens of thousands but perhaps millions – even optimizing our noisy estimate of the uncertainty coverage does not introduce substantial error.[4] If $U$ is given by the margin between probabilistic predictions, then $U_{\max} \le 1$; other notions can also be easily bounded. Assuming $k_\sigma \le 1$ is for convenience, but a different upper bound can simply be absorbed into $U_{\max}$. The bound is indeed 1 for the Gaussian kernel on $g$ used by Bae et al. (2024) (as well as by us); this kernel also has $L_\sigma = \sqrt{\frac{2}{e}} \cdot \frac{1}{\sigma}$. The kernel which corresponds to the probability coverage of Yehuda et al. (2022), however, is not Lipschitz, suggesting why it is so sensitive to $\sigma$. Self-supervised representations in $g$ are often normalized to $R = 1$, and usually have $d$ at most a few hundred.

---

[3]$k_\sigma(\mathbf{x},\mathbf{x}';g)$ is in the form of $\psi((g(\mathbf{x})-g(\mathbf{x}'))/\sigma) \in [0,1]$ for some function $\psi$ and a feature mapping $g$.

[4]It is worth emphasizing that, although Bae et al. (2024) mentioned a simple Hoeffding bound for $\widehat{\text{C}}_{k_\sigma}$, this bound only applies to fixed $\mathcal{S}$ *independent of* $\mathcal{U}$ – while in reality $\mathcal{S} \subseteq \mathcal{U}$. Ignoring this problem and taking a union bound over all $\binom{N}{B}$ subsets of $\mathcal{U}$ would yield a uniform convergence bound of $U_{\max}\sqrt{\frac{1}{2N}\log\left(\binom{N}{B}\frac{2}{\delta}\right)} \le U_{\max}\sqrt{\frac{B}{2N}\left[\log\left(e\frac{N}{B}\right) + \frac{1}{B}\log\frac{2}{\delta}\right]}$. The rate of Theorem 2 is very similar, with the advantage of being correct.

---

**Algorithm 1:** Uncertainty herding with parameter adaptation

---

**Input:** Initial labeled set $\mathcal{L}_0$, Initial unlabeled set $\mathcal{U}_0$, a set of temperatures $\mathcal{T}$, the number of iterations T, a set of query budgets $\{B_t\}_{t=0}^{T-1}$, a classifier $f$, and a feature extractor $g$

1 **for** $t \in [0, 1, \cdots, T-1]$ **do**

    // Parameter adaptation

2      Compute $\tau^* = \arg\min_{\tau \in \mathcal{T}} \text{ECE}(f^{\mathcal{L}_t^{\text{train}}}, \mathcal{L}_t^{\text{val}})$ where $\mathcal{L}_t^{\text{train}}$ and $\mathcal{L}_t^{\text{val}}$ are random split from $\mathcal{L}_t$, and $f^{\mathcal{L}_t^{\text{train}}}$ refers to a classifier $f$ trained on $\mathcal{L}_t^{\text{train}}$

3      Compute $\mathbf{k} \in \mathbb{R}^{|\mathcal{U}_t|}$ with $\mathbf{k}_n = \max_{\mathbf{x}' \in \mathcal{L}_t} k_{\sigma^*}(\mathbf{x}_n, \mathbf{x}')$ for $\sigma^* = \min_{\mathbf{u}, \mathbf{v} \in \mathcal{L}_t, \mathbf{u} \neq \mathbf{v}} D(\mathbf{u}, \mathbf{v}; g)$

    // Greedy selection based on the uncertainty coverage

4      **for** $b \in [1, 2, \cdots, B_t]$ **do**

5          Select $\mathbf{x}_b^* = \arg\max_{\tilde{\mathbf{x}} \in \mathcal{U}} \frac{1}{N} \sum_{n=1}^N U(x_n; f_{\tau^*}) \cdot \max(k_{\sigma^*}(\mathbf{x}_n, \tilde{\mathbf{x}}) - \mathbf{k}_n, 0)$

6          Update $\mathbf{k}_n \leftarrow \max(k_{\sigma^*}(\mathbf{x}_n, \mathbf{x}_b^*), \mathbf{k}_n), \forall n \in |\mathcal{U}_t|$

7      Update $\mathcal{L}_{t+1} \leftarrow \mathcal{L}_t \cup \{\mathbf{x}_b^*\}_{b=1}^{B_t}$ and $\mathcal{U}_{t+1} \leftarrow \mathcal{U}_t \setminus \{\mathbf{x}_b^*\}_{b=1}^{B_t}$

---

## 3.2 PARAMETER ADAPTATION

We wish to choose parameters of UCoverage such that it smoothly changes from behaving like generalized coverage in low-budget regimes to behaving like uncertainty in high-budget regimes.

**Handling the low-budget case: calibration**    When $|\mathcal{S}|$ is small, we would like to roughly replicate GCoverage. We can do this by making our uncertainty function constant:

**Proposition 3.** *If $\forall \mathbf{x} \in \mathcal{U}, U(\mathbf{x}; f) \to c$ where $c \geq 0$, the estimated UCoverage $\widehat{\text{UC}}_{k_\sigma}(\mathcal{S})$ approaches the estimated GCoverage $\widehat{\text{C}}_{k_\sigma}(\mathcal{S})$, up to a constant.*

When $|\mathcal{L}|$ is very small, our model $f$ is bad. Models with poor predictive power will generally have near-constant uncertainty if they are well-*calibrated*. We thus encourage calibration primarily through temperature scaling, a simple but effective post-hoc calibration method (Guo et al., 2017):

1. Split $\mathcal{L}_t$ into $\mathcal{L}_t^{\text{train}}$ and $\mathcal{L}_t^{\text{val}}$, and train a model $f$ on $\mathcal{L}_t^{\text{train}}$, obtaining $f^{\mathcal{L}_t^{\text{train}}}$.
2. Choose $\tau^*$ among some candidate set $\mathcal{T}$ to minimize the expected calibration error (ECE)[5] (Naeini et al., 2015) of the temperature-scaled predictions $f(\mathbf{x})/\tau$ on $\mathcal{L}_t^{\text{val}}$. where $f(\mathbf{x}) \in \mathbb{R}^K$ denotes a logit vector with $K$ being the number of classes.
3. Compute uncertainties with $\tau^*$-scaled softmax: $\log \hat{p}_{\tau^*}(\mathbf{x}) \propto f_{\tau^*}(\mathbf{x}) := f(\mathbf{x})/\tau^*$.

The selected temperatures $\tau^*$ are generally large when $|\mathcal{L}|$ is small and so $f$ has poor predictions, which makes $U(\mathbf{x}; f)$ close to constant. As $|\mathcal{L}|$ increases and $f$'s predictions improve, $\tau^*$ decreases (see Figure 1b, left panel), making uncertainty values more distinct.

**Handling the high-budget case: decreasing $\sigma$**    As $|\mathcal{L}|$ increases, the effect of a single new data point on the trained model tends to become more "semantically local," implying it is reasonable to treat points as "covering" only closer and closer points in $g$ space by decreasing the radius $\sigma$. As a heuristic, we choose the radius to be the minimum distance between data points in the labeled set $\mathcal{L}$: $\sigma^* = \min_{\mathbf{u}, \mathbf{v} \in \mathcal{L}_t, \mathbf{u} \neq \mathbf{v}} \|g(\mathbf{u}) - g(\mathbf{v})\|$. Since $\mathcal{U}$ is bounded, as $|\mathcal{L}|$ grows we have that $\sigma^* \to 0$ (see Figure 1b, right panel); thus Proposition 4 eventually applies, becoming uncertainty-based selection.

**Proposition 4.** *Suppose $k_\sigma(\mathbf{x}, \mathbf{x}'; g) = \psi((g(\mathbf{x}) - g(\mathbf{x}'))/\sigma)$ for a fixed $g : \mathcal{X} \to \mathbb{R}^d$ which is injective on $\mathcal{U}$ and a function $\psi : \mathbb{R}^d \to [0, 1]$ with $\psi(0) = 1$ and for all $t \in \mathbb{R}^d$ with $\|t\| = 1$, $\lim_{a \to \infty} \psi(at) = 0$. If $\sigma \to 0$, the estimated uncertainty coverage $\widehat{\text{UC}}_{k_\sigma}(\mathcal{S})$ approaches $\sum_{s=1}^{|\mathcal{S}|} U(\mathbf{x}_s; f)$, up to a constant.*

As we shall see in Section 4.4, UCoverage with fixed $\tau$ and $\sigma$ is not robust across budget levels, performing worse than MaxHerding in low-budget and worse than Margin in high-budget regimes. With our adaption techniques, however, UCoverage outperforms competitors across label budgets.

---

[5]$\text{ECE}(f, \mathcal{D}) = \frac{1}{|\mathcal{D}|} \sum_{m=1}^M |B_m| \cdot |\text{Acc}(B_m, f) - \text{Conf}(B_m, f)|$ measures the difference between actual outcomes (Acc) and confidence (Conf) for a model $f$ and dataset $\mathcal{D}$. $B_m$ is a set of samples falling in bin $m$.

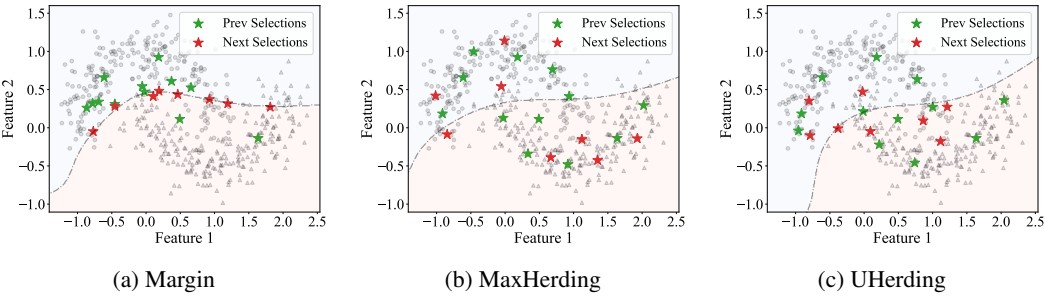

(a) Margin            (b) MaxHerding            (c) UHerding

Figure 2: Comparison of Margin, MaxHerding and proposed UHerding on half-moon toy data.

### 3.3 UNCERTAINTY HERDING

To obtain an actual active learning method, we still need an algorithm to maximize $\widehat{\mathrm{UC}}_{k_\sigma}$. We could select a batch by finding $\bar{\mathcal{S}} \in \arg\max_{\mathcal{S}\subseteq\mathcal{U}, |\mathcal{S}|=B} \widehat{\mathrm{UC}}_{k_\sigma}(\mathcal{L} \cup \mathcal{S})$. This is equivalent to the weighted kernel $k$-medoids objective, with weights determined by uncertainty $U(\mathbf{x}; f)$; thus, we could use Partitioning Around Medoids (PAM) (Kaufman & Rousseeuw, 2009) to try to find $\bar{\mathcal{S}}$.

Bae et al. (2024), however, observed that even with a highly optimized implementation, this algorithm is much slower to optimize GCoverage than greedy methods, with little improvement in active learning performance. We thus focus on greedy selection, which we call Uncertainty Herding (UHerding) by analogy to MaxHerding (itself an analogy to kernel herding, Chen et al., 2010).

**Definition 5** (Uncertainty Herding). To greedily add a single data point to a set $\mathcal{L}' = \mathcal{L} \cup \mathcal{S}$, select

$$\mathbf{x}^* \in \arg\max_{\tilde{\mathbf{x}}\in\mathcal{U}} \widehat{\mathrm{UC}}_{k_\sigma}(\mathcal{L}' \cup \{\tilde{\mathbf{x}}\}) = \arg\max_{\tilde{\mathbf{x}}\in\mathcal{U}} \left( \frac{1}{N} \sum_{n=1}^{N} U(\mathbf{x}_n; f_\tau) \cdot \max_{\mathbf{x}'\in\mathcal{L}'\cup\{\tilde{\mathbf{x}}\}} k_\sigma(\mathbf{x}_n, \mathbf{x}'; g) \right). \quad (2)$$

UHerding selects a new batch $\mathcal{S}$ of size $|\mathcal{S}|$ by picking one point at a time to add to $\mathcal{L}'$.

UHerding improves uncertainty measures by accounting for both the uncertainty of a selected point and its influence on reducing nearby uncertainty. It improves on MaxHerding by incorporating uncertainty, putting less weight on covering already-certain points.

**Corollary 6.** *In the setting of Theorem 2, let $\hat{\mathcal{S}} \subseteq \mathcal{U}$ be the result of UHerding for $B$ steps to add to $\mathcal{L}$, and $\mathrm{UC}^* = \max_{\mathcal{S}\subseteq\mathcal{U}, |\mathcal{S}|=B} \mathrm{UC}_{k_\sigma}(\mathcal{L} \cup \mathcal{S})$ the optimal coverage obtainable among $\mathcal{U}$. Then*

$$\mathrm{UC}_{k_\sigma}(\mathcal{L} \cup \hat{\mathcal{S}}) \geq \left(1 - \frac{1}{e}\right) \mathrm{UC}^* - \left(2 - \frac{1}{e}\right) U_{\max} \sqrt{\frac{B}{N}} \left[ 8L_\sigma + \frac{1}{2}\sqrt{d\log\left(R^2\frac{N}{B}\right) + \frac{2}{B}\log\frac{2}{\delta}} \right].$$

Figure 2 visually compares the next selected data points (in red) by Margin, MaxHerding, and UHerding (using Margin uncertainty) on a two-class half-moon dataset represented by ○ and △. A logistic regression model with fifth-order polynomial features is trained at each iteration, with the decision boundary after training on 12 previously selected points (in green) shown as dashed lines. As expected, Margin selects points near the decision boundary, which can lead to suboptimal models if the predicted boundary deviates from the true one. MaxHerding selects the most representative points based on prior selections but ignores model predictions; this gives quick generalization with few labeled points, but performance saturates over time as it neglects points near the boundary. UHerding balances these approaches by initially selecting representative points and gradually focusing on uncertain points near the boundary, improving performance over time.

### 3.4 CONNECTION TO HYBRID METHODS

UHerding is closely connected to existing hybrid active learning methods. As mentioned in Section 2, we replace $k$-means and $k$-means++ in various algorithms with greedy kernel $k$-medoids (MaxHerding) to simplify our arguments; this does not harm their effectiveness. We also apply a self-supervised feature extractor $g$ instead of a feature extractor from a classifier $f$, since feature

embeddings from $g$ are more informative when $f$ is trained on a small labeled set. Finally, we often assume that $k_\sigma$ is in fact a positive-definite (RKHS) kernel; this is true for the kernels we use.

**Proposition 7** (Weighted $k$-means of Zhdanov 2019). *Define an uncertainty measure $U(\mathbf{x}; f)$ from another uncertainty measure $U(\mathbf{x}; f)$ as $U(\mathbf{x}; f) := U'(\mathbf{x}; f) \cdot \mathbb{1}[U'(\tilde{\mathbf{x}}; f) \geq \nu]$, where $\nu \geq 0$ satisfies $\sum_{n=1}^{N} \mathbb{1}[U'(\mathbf{x}_n; f) \geq \nu] = M$, a pre-defined number. Then weighted $k$-means with uncertainty $U'$, changed to use greedy kernel $k$-medoids, is UHerding with uncertainty $U$ and the same kernel.*

As we shall see in Section 4, UHerding significantly outperforms weighted $k$-means, indicating that it is crucial to (a) convert $k$-means into MaxHerding and (b) apply parameter adaptation.

**Proposition 8** (ALFA-Mix of Parvaneh et al. 2022). *Let $\hat{y}(\cdot; f)$ be the predicted label of an input under $f$. Define an uncertainty measure*

$$U(\mathbf{x}; f) := \mathbb{1}\left[\exists \ class \ j \ s.t. \ \hat{y}\big(\alpha_j(\mathbf{x}) \, g(\mathbf{x}) + (1 - \alpha_j(\mathbf{x})) \, \bar{g}_j; f\big) \neq \hat{y}(g(\mathbf{x}); f)\right] \quad (3)$$

*where $\bar{g}^j$ is the mean of feature representations belonging to class $j$ and $\alpha_j(\mathbf{x}) \in [0, 1)$ is the same parameter as determined by ALFA-Mix. Then ALFA-Mix, with clustering replaced by greedy kernel $k$-medoids, is UHerding with uncertainty $U$ and the same kernel.*

The equivalence of weighted $k$-means and ALFA-Mix to UHerding with the right choice of uncertainty measure implies that Propositions 3 and 4 also apply to them. There is a weaker connection to BADGE; UHerding and BADGE are not equivalent with any choice of uncertainty measure. However, a variant of BADGE with greedy kernel $k$-medoids uses the kernel,

$$h(\mathbf{x}_n, \mathbf{x}') = 2\langle q(\mathbf{x}_n), q(\mathbf{x}')\rangle k_\sigma(\mathbf{x}_n, \mathbf{x}'; g) - \|q(\mathbf{x}_n)\|_2^2 \, k_\sigma(\mathbf{x}_n, \mathbf{x}_n; g) - \|q(\mathbf{x}')\|_2^2 \, k_\sigma(\mathbf{x}', \mathbf{x}'; g) \quad (4)$$

where $q(\mathbf{x}) = \hat{y}(\mathbf{x}; f) - \hat{p}(\mathbf{x}; f)$ with $f$ being a classifier. This does satisfy the following statement, showing properties similar to Propositions 3 and 4 hold.

**Proposition 9** (BADGE, Ash et al., 2020). *If $\forall \mathbf{x} \in \mathcal{U}$, $\hat{p}(\mathbf{x}; f) \to \frac{1}{K}\vec{1}$, then this BADGE approaches a slightly modified MaxHerding: $\left(\mathbb{1}\left[\hat{y}(\mathbf{x}_n; f) = \hat{y}(\mathbf{x}'; f)\right] - \frac{1}{K}\right) k_\sigma(\mathbf{x}_n, \mathbf{x}'; g)$ instead of $k_\sigma(\mathbf{x}_n, \mathbf{x}'; g)$. If $\sigma \to 0$, it approaches to the uncertainty-based method where uncertainty is defined as, $U''(\tilde{\mathbf{x}}) := \min_{\mathbf{x}' \in \mathcal{L} \cup \{\tilde{\mathbf{x}}\}} \|\hat{y}(\mathbf{x}'; f) - \hat{p}(\mathbf{x}'; f)\|_2^2$.*

Using the parameter adaptation, hybrid methods can also smoothly interpolate between MaxHerding and uncertainty; Figure 6b shows this improves BADGE. Although selecting data points maximizing $U''(\tilde{\mathbf{x}})$ is somewhat counter-intuitive, lowering $\sigma$ still helps as it does not go too close to 0.

## 4 EXPERIMENTS

In this section, we assess the robustness of the proposed UHerding against existing active learning methods for standard supervised learning (Sections 4.1 and 4.2) and transfer learning (Section 4.3) across several benchmark datasets: CIFAR10 (Krizhevsky, 2009), CIFAR100 (Krizhevsky et al.), TinyImageNet (mnmoustafa, 2017), ImageNet (Deng et al., 2009), and DomainNet (Peng et al., 2019). Section 4.4 gives ablation studies to see how each component of UHerding contributes.

**Active learning methods** We compare with several active learning methods listed below. We exclude ProbCover; its generalization MaxHerding reliably outperforms it (Bae et al., 2024).

*Random* Uniformly select random $B$ data points from $\mathcal{U}$.

*Confidence* Iteratively select $\mathbf{x}^* \in \arg\min_{\tilde{\mathbf{x}} \in \mathcal{U}} p_1(y|\tilde{\mathbf{x}})$, with $p_1$ the highest predicted probability.

*Margin (Scheffer et al., 2001)* Iteratively select $x^* = \arg\min_{\tilde{\mathbf{x}} \in \mathcal{U}} p_1(y|\tilde{\mathbf{x}}) - p_2(y|\tilde{\mathbf{x}})$, where $p_2$ is the second-highest predicted probability.

*Entropy (Wang & Shang, 2014)* Iteratively select $\mathbf{x}^* = \arg\max_{\tilde{\mathbf{x}} \in \mathcal{U}} \mathrm{H}(\hat{y}(\tilde{\mathbf{x}}) \mid \tilde{\mathbf{x}})$, where $\mathrm{H}(\cdot)$ is the Shannon entropy.

*Weighted Entropy (Zhdanov, 2019)* Select data points closest to the centroids of weighted $k$-means using unlabeled data points with high enough margins (as in Margin selection above).

*BADGE (Ash et al., 2020)* Select points with the $k$-means++ initialization algorithm using gradient embeddings w.r.t. the weights of the last layer.

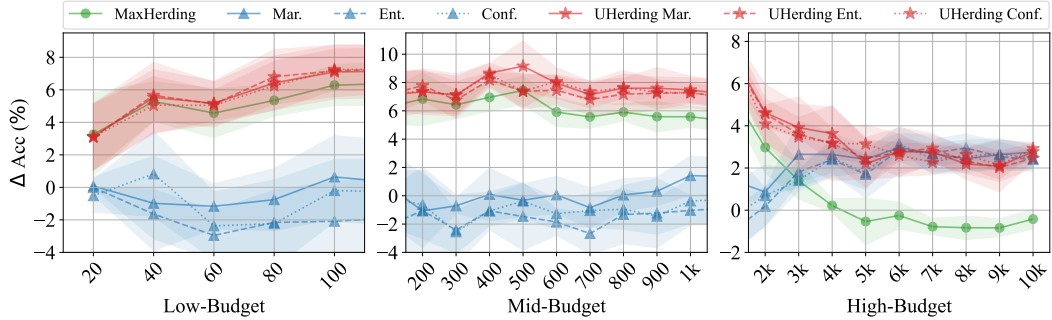

Figure 3: UHerding versus MaxHerding and uncertainty, with different uncertainty measures. Mean and standard deviation of 5 runs of the difference between a method and Random selection.

*ALFA-Mix (Parvaneh et al., 2022)* Select data points closest to the centroids of $k$-means only using uncertain unlabeled data points, based on feature interpolation.

*ActiveFT (Xie et al., 2023)* Select data points close to parameterized "centeroids," learned by minimizing KL between the unlabeled set and selected set, along with diversity regularization.

*Typiclust (Hacohen et al., 2022)* Run a clustering algorithm, *e.g.*, $k$-means. For each cluster, select a point with the highest "typicality" using $m$-Nearest Neighbors.

*Coreset (Sener & Savarese, 2018)* Iteratively select points with the $k$-Center-Greedy algorithm: $\mathbf{x}^* = \arg\max_{\tilde{\mathbf{x}} \in \mathcal{U}} \min_{\mathbf{x}' \in \mathcal{L}} \|\tilde{\mathbf{x}} - \mathbf{x}'\|_2$.

*MaxHerding (Bae et al., 2024)* Iteratively select points to maximize the generalized coverage (1).

**Implementation** We always re-initialize models (cold-start), randomly or from pre-trained parameters, after each round of acquisition, rather than warm-starting from the previous model (Ash & Adams, 2020). We manually categorize the budget regimes into low, mid, and high for supervised learning, and low and high for transfer learning tasks. Representation-based methods win in low-budget regimes, while uncertainty-based begin to catch up in mid-budget, and win in high-budget.

### 4.1 UHERDING INTERPOLATES, AND THE UNCERTAINTY MEASURE DOESN'T MATTER

We first aim to verify that UHerding effectively interpolates between MaxHerding and uncertainty measures across budget regimes. We train a randomly-initialized ResNet18 (He et al., 2016) on CIFAR10 using 5 random seeds, gradually increasing the size of the labeled set, as shown in Figure 3. We consider UHerding based on Margin (Mar.), Entropy (Ent.), and Confidence (Conf.). The y-axis of Figure 3 represents $\Delta$Acc, indicating the performance difference relative to Random selection.

UHerding performs slightly better or comparably to MaxHerding in the low-budget regime, with a growing performance gap in the mid-budget regime, and substantial improvements in the high-budget regime. The opposite is true with uncertainty measures: UHerding is substantially better with low budgets, while they eventually catch up and tie UHerding with high budgets.

The choice of uncertainty measure has little impact on performance. We thus only use Margin as the UHerding uncertainty measure in the future unless otherwise specified.

### 4.2 COMPARISON WITH STATE OF THE ART

We now compare UHerding with state-of-the-art active learning methods, particularly hybrids, to assess their robustness across different budget regimes. Figure 4 compares CIFAR100 and TinyImagenet, using 3 runs of a randomly initialized ResNet18; CIFAR10 results are in Appendix B.

Overall trends are similar to before: representation-based methods are good with low budgets but lose as the budget increases; uncertainty-based and hybrid methods are largely the opposite. UHerding with Margin uncertainty (UHerding Mar.) wins convincingly: no competitor ever outperforms UHerding, and for each competitor there is some budget where UHerding wins substantially.

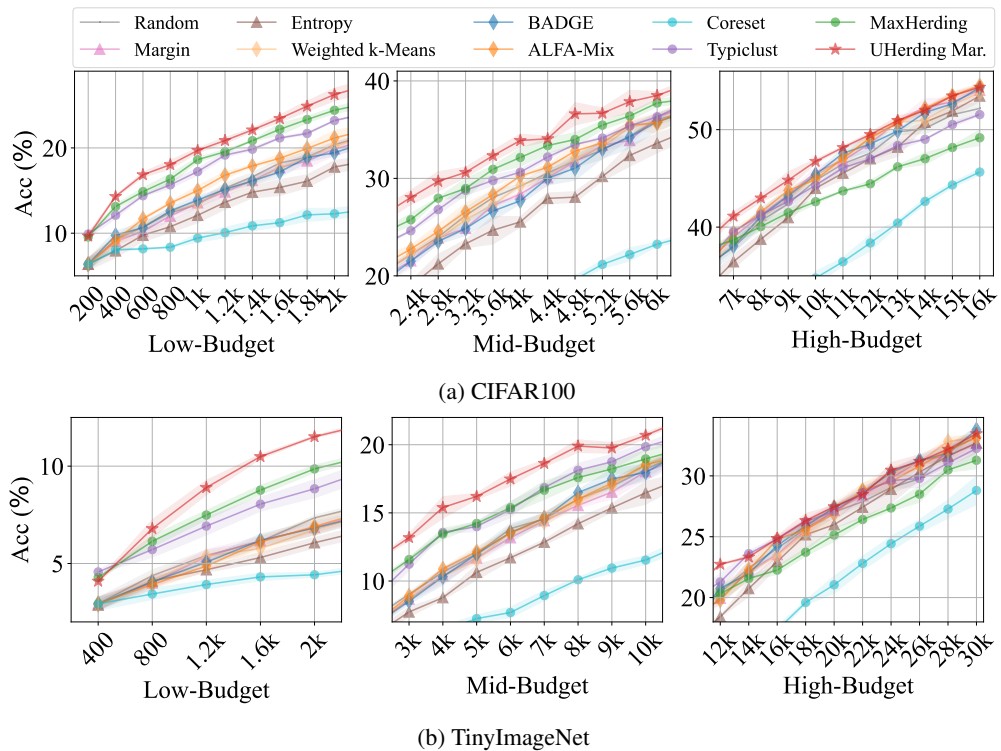

Figure 4: Comparison on CIFAR100 and TinyImageNet for supervised-learning tasks.

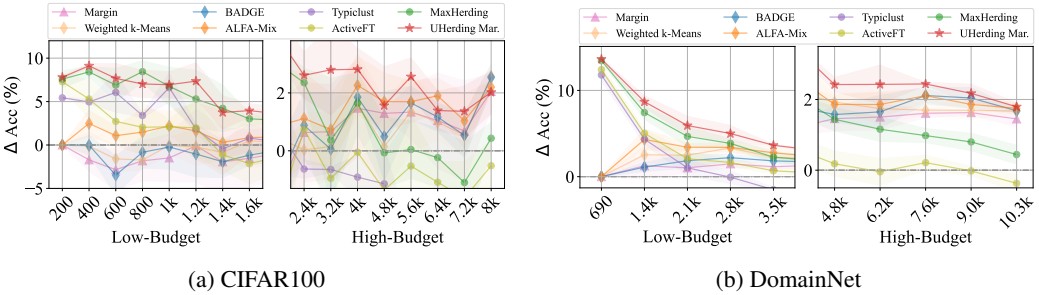

Figure 5: Comparison on CIFAR100 and DomainNet for transfer learning tasks.

### 4.3 COMPARISON FOR TRANSFER LEARNING TASKS

Fine-tuning foundation models to new tasks or datasets is of increasing importance. Inspired by ActiveFT (Xie et al., 2023), we compare UHerding to leading approaches for active transfer learning.

We use DeiT (Touvron et al., 2021) pre-trained on ImageNet (Deng et al., 2009), following Parvaneh et al. (2022); Xie et al. (2023). We fine-tune the entire model, using DeiT Small for CIFAR-100 and DeiT Base for DomainNet. Figure 5 compares UHerding with various active learning methods, including ActiveFT, which consistently underperforms the other methods likely due to its design being optimized for single-iteration data selection. On CIFAR100, UHerding is comparable to representation-based methods in the low-budget regime but surpasses them by about 2% in the high-budget; it substantially outperforms uncertainty and hybrid methods in the low-budget regime and ties or wins with high-budgets. On DomainNet, UHerding outperforms MaxHerding by 1.5–2% even in the low-budget. Compared to the best-performing hybrid method ALFA-Mix, UHerding wins by up to 13% with low budgets and performs similarly with high budgets.

It is also common to fine-tune only the last few layers, especially in meta-learning (Wang et al., 2019; Chen et al., 2019; Goldblum et al., 2020) and self-supervised learning (Chen et al., 2020; Caron et al., 2021). Similarly to Bae et al. (2024), we use DINO (Caron et al., 2021) features fixed through fine-

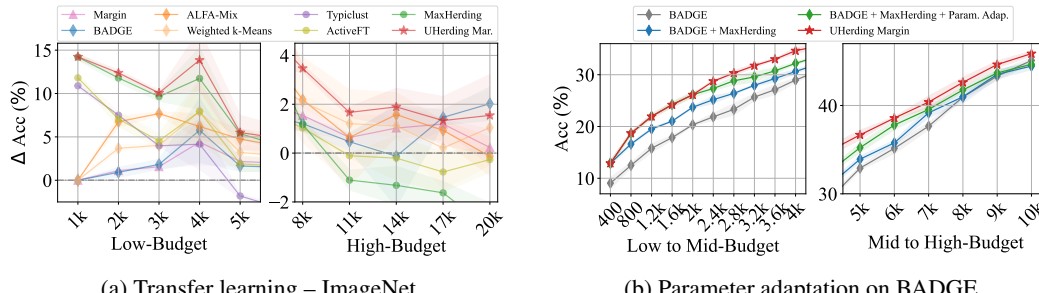

(a) Transfer learning – ImageNet          (b) Parameter adaptation on BADGE

Figure 6: (Left) results on ImageNet for fine-tuning, (Right) application of parameter adaptation.

| Method | Low | | | | | Middle | | | High | | | | |
|--------|-----|------|-------|------|------|--------|------|-------|------|------|-------|------|------|
|        | C10 | C100 | Tiny. | Dom. | ImN. | C10 | C100 | Tiny. | C10 | C100 | Tiny. | Dom. | ImN. |
| Entropy | -1.8 | -1.7 | -0.6 | -0.2 | 1.7 | -1.6 | -2.9 | -1.7 | 2.2 | -0.6 | -0.7 | 0.7 | **1.2** |
| Margin | -0.4 | -0.3 | -0.2 | 1.0 | 1.8 | -0.1 | -0.4 | -0.3 | **2.5** | **1.1** | 0.0 | 1.5 | 0.9 |
| BADGE | -0.5 | -0.1 | -0.2 | 1.4 | 2.0 | 0.6 | -0.7 | 0.0 | 2.2 | 0.9 | **0.4** | **1.8** | **1.0** |
| ALFA-M | 0.1 | 0.9 | -0.3 | 2.8 | 5.1 | 1.1 | 0.6 | 0.1 | **2.3** | **1.3** | 0.2 | **1.9** | **1.0** |
| Weight. k | -0.5 | -0.1 | -0.3 | 2.1 | 3.8 | 0.9 | 0.0 | -0.2 | 1.8 | 0.8 | **0.3** | 1.7 | 0.8 |
| Coreset | -2.7 | -4.5 | -1.4 | -3.5 | -6.6 | -13 | -11 | -5.4 | -10 | -9.6 | -5.5 | -2.7 | -12 |
| ActiveFT | – | – | – | **4.4** | **6.6** | – | – | – | – | – | – | 0.0 | -0.1 |
| Typiclust | **3.7** | **3.3** | **1.6** | 3.1 | 4.9 | **4.9** | **1.8** | **2.1** | -0.8 | -0.1 | **0.3** | -3.2 | -9.9 |
| MaxHerd. | **5.0** | **4.1** | **2.1** | **6.2** | **10.6** | **6.2** | **2.8** | 1.9 | 0.1 | -2.2 | -1.5 | 1.0 | -1.2 |
| UHerding | **5.5** | **5.5** | **3.1** | **7.4** | **11.2** | **7.8** | **4.3** | **3.7** | **3.0** | **2.1** | **0.8** | **2.3** | **2.0** |

Table 1: Comparison of the mean improvement/degradation over Random selection on each budget regime and dataset. The **first**, **second**, **third** best results for each setting are marked.

tuning; we train a head of three fully connected ReLU layers on ImageNet. Figure 6a shows similar results, with UHerding consistently outperforming other methods across various budget regimes.

Table 1 summarizes the results of Sections 4.2 and 4.3 and Appendix B, reporting the mean improvement/degradation over Random for each budget regime.[6] UHerding wins across all budget regimes, while other methods have a significant range of budgets where they are worse than Random.

### 4.4 ABLATION STUDY

Figure 6b gradually modifies BADGE to be similar to UHerding with Margin uncertainty on CIFAR-100, replacing $k$-means++ with MaxHerding, then adding parameter adaption. Each step improves; so does the final step to UHerding, which changes the choice of uncertainty.

In Appendix C, we analyze the contributions of each parameter adaptation component in UHerding. The baseline with fixed parameters performs slightly worse than MaxHerding at low- and mid-budgets, and worse than Margin at high-budgets. Adding temperature scaling results in notable improvements across all budget regimes, with further gains from incorporating radius adaptation.

### 5 CONCLUSION

In this work, we introduced uncertainty coverage, an objective that unifies low- and high-budget active learning objectives through a smooth interpolation with adaptive parameter adjustments. We showed generalization guarantees for the optimization of this coverage. By identifying conditions under which uncertainty coverage approaches generalized coverage and uncertainty measures, we made UHerding robust across various budget regimes. This adaptation also enhances an existing hybrid active learning method when similar parameter adjustments are applied. UHerding achieves state-of-the-art performance across various active learning and transfer learning tasks and, to our knowledge, is the only method that consistently performs well in both low- and high-budget settings.

---

[6]ActiveFT is reported only for DomainNet (Dom.) and ImageNet (ImN.), as it is designed for fine-tuning.

ACKNOWLEDGMENTS

This work was supported in part by Mitacs through the Mitacs Accelerate program, the Natural Sciences and Engineering Resource Council of Canada, the Canada CIFAR AI chairs program, and Advanced Research Computing at the University of British Columbia.

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

# A  PROOFS

## A.1  ESTIMATION QUALITY

**Theorem 2.** *Let* $U(\mathbf{x}; f) \in [0, U_{\max}]$, $k_\sigma(\mathbf{x}, \mathbf{x}'; g) = \tilde{k}_\sigma(g(\mathbf{x}), g(\mathbf{x}')) \in [0, 1]$, $\{g(\mathbf{x}) : \mathbf{x} \in \mathcal{U}\} \subseteq$ $\{\mathbf{t} \in \mathbb{R}^d : \|\mathbf{t}\| \leq R\}$, *and* $\left| \tilde{k}_\sigma(\mathbf{t}, \mathbf{t}') - \tilde{k}_\sigma(\mathbf{t}, \mathbf{t}'') \right| \leq L_\sigma \|\mathbf{t}' - \mathbf{t}''\|$. *Let* $\mathcal{L} \subseteq \mathcal{X}$ *be arbitrary and fixed. Assume* $B/N < 16R^2$. *Then, with probability at least* $1 - \delta$ *over the choice of the* $N$ *iid data points in* $\mathcal{U} \subseteq \mathcal{X}$ *used to estimate* $\widehat{\mathrm{UC}}_{k_\sigma}$, *all size-*$B$ *sets* $\mathcal{S}$ *(not only subsets of* $\mathcal{U}$*) have low error:*

$$\sup_{\substack{\mathcal{S} \subseteq \mathcal{X} \\ |\mathcal{S}| = B}} \left| \mathrm{UC}_{k_\sigma}(\mathcal{L} \cup \mathcal{S}) - \widehat{\mathrm{UC}}_{k_\sigma}(\mathcal{L} \cup \mathcal{S}) \right| \leq U_{\max} \sqrt{\frac{B}{N}} \left[ 8 L_\sigma + \frac{1}{2} \sqrt{d \log \left( R^2 \frac{N}{B} \right) + \frac{2}{B} \log \frac{2}{\delta}} \right].$$

*Proof.* Rather than operating directly on sets $\mathcal{S} \subseteq \mathcal{X}$, we will operate on $B$-tuples $\mathcal{T} \in (\mathbb{R}^d)^B$ corresponding to $(g(\mathbf{s}_1), \ldots, g(\mathbf{s}_B))$. As these are ordered tuples, the mapping from $\mathcal{T}$ to $\mathcal{S}$ is many-to-one even if $g$ is injective; if we prove convergence for all $\mathcal{T}$, it will necessarily prove convergence for all $\mathcal{S}$. Define $F(\mathcal{T}) = \mathrm{UC}_{k_\sigma}(\mathcal{L} \cup \mathcal{S})$ and $\hat{F}(\mathcal{T}) = \widehat{\mathrm{UC}}_{k_\sigma}(\mathcal{L} \cup \mathcal{S})$ for any $\mathcal{S}$ corresponding to that $\mathcal{T}$; this is well-defined, since $\mathrm{UC}_{k_\sigma}$ depends on $\mathcal{S}$ only through $\{g(\mathbf{s}) : \mathbf{s} \in \mathcal{S}\}$.

Now, we will construct a vector space for elements $\mathcal{T} = (\mathbf{t}_1, \ldots, \mathbf{t}_B)$. Vector addition and scalar multiplication are defined elementwise, and a norm is defined as $\|\mathcal{T}\| = \max(\|\mathbf{t}_1\|, \ldots, \|\mathbf{t}_B\|)$, where $\|\mathbf{t}_i\|$ is the standard Euclidean norm. This space is complete, and hence a Banach space of dimension $Bd$.

The uncertainty coverage $F(\mathcal{T})$ is Lipschitz with respect to the Banach space norm:

$$\begin{aligned}
|F(\mathcal{T}) - F(\mathcal{T}')| &\leq \mathbb{E}_\mathbf{x} U(\mathbf{x}; f) \left| \max_{\mathbf{t} \in \{g(\tilde{\mathbf{x}}) : \tilde{\mathbf{x}} \in \mathcal{L}\} \cup \mathcal{T}} \tilde{k}_\sigma(g(\tilde{\mathbf{x}}), \mathbf{t}) - \max_{\mathbf{t}' \in \{g(\tilde{\mathbf{x}}) : \tilde{\mathbf{x}} \in \mathcal{L}\} \cup \mathcal{T}'} \tilde{k}_\sigma(g(\mathbf{x}), \mathbf{t}') \right| \\
&\leq \mathbb{E}_\mathbf{x} U(\mathbf{x}; f) \max_{i \in [B]} \left| \tilde{k}_\sigma(g(\mathbf{x}), \mathbf{t}_i) - \tilde{k}_\sigma(g(\mathbf{x}), \mathbf{t}'_i) \right| \\
&\leq \mathbb{E}_\mathbf{x} U(\mathbf{x}; f) \max_{i \in [B]} L_\sigma \|\mathbf{t}_i - \mathbf{t}'_i\| \\
&= L_\sigma \left( \mathbb{E}_{\tilde{\mathbf{x}}} U(\tilde{\mathbf{x}}; f) \right) \|\mathcal{T} - \mathcal{T}'\|.
\end{aligned}$$

The second inequality holds because the maximum function is Lipschitz on $\mathbb{R}^N$. Specifically, let $a_1, \ldots, a_N, a'_1, \ldots, a'_N \in \mathbb{R}$, and let $\hat{n} \in \arg\max_{n \in [N]} a_n$. Then

$$\max_n a_n - \max_{n \in [N]} a'_n = a_{\hat{n}} - \max_{n \in N} a'_n \leq a_{\hat{n}} - a'_{\hat{n}} \leq \max_{n \in N} a_n - a'_n \leq \max_{n \in N} |a_n - a'_n|,$$

and, symmetrically, it is at least $-\max |a_n - a'_n|$, so $|\max_n a_n - \max_n a'_n| \leq \max_n |a_n - a'_n|$.

As $\hat{F}$ is exactly $F$ with an empirical distribution for $\mathbf{x}$, it is $L_\sigma \left( \frac{1}{N} \sum_{n=1}^N U(\mathbf{x}_n) \right)$-Lipschitz. We thus have

$$\begin{aligned}
\left| \left( F(\mathcal{T}) - \hat{F}(\mathcal{T}) \right) - \left( F(\mathcal{T}') - \hat{F}(\mathcal{T}') \right) \right| &\leq L_\sigma \left( \mathbb{E}_\mathbf{x} U(\mathbf{x}) + \frac{1}{N} \sum_{n=1}^N U(\mathbf{x}) \right) \|\mathcal{T} - \mathcal{T}'\| \\
&\leq 2 L_\sigma U_{\max} \|\mathcal{T} - \mathcal{T}'\|.
\end{aligned}$$

By Proposition 5 of Cucker & Smale (2001), we can cover the ball $\{\mathcal{T} : \|\mathcal{T}\| \leq R\}$ with at most $(4R/\eta)^{Bd}$ balls of radius $\eta$ with respect to the metric $\|\mathcal{T} - \mathcal{T}'\|$. So, to construct our covering argument, we apply the (bidirectional) Hoeffding inequality to the center of each of these balls, with failure probability $\delta / (4R/\eta)^{Bd}$ for each. Combining this with how much $F - \hat{F}$ can change between an arbitrary point in $\{\mathcal{T} : \|\mathcal{T}\| \leq R\}$ and the nearest center, we have that for all $\eta \in (0, R)$, it holds with probability at least $1 - \delta$ that

$$\sup_{\mathcal{T}} \left| F(\mathcal{T}) - \hat{F}(\mathcal{T}) \right| \leq 2 L_\sigma U_{\max} \eta + U_{\max} \sqrt{\frac{Bd}{2N} \log \frac{4R}{\eta} + \frac{1}{2N} \log \frac{2}{\delta}}.$$

The result follows by picking $\eta = 4\sqrt{B/N}$ and using $\log(a) = \frac{1}{2} \log(a^2)$. $\qquad\square$

## A.2 Parameter Adaptation

**Proposition 3.** *If $\forall \mathbf{x} \in \mathcal{U}, U(\mathbf{x}; f) \to c$ where $c \geq 0$, the estimated UCoverage $\widehat{\mathrm{UC}}_{k_\sigma}(\mathcal{S})$ approaches the estimated GCoverage $\widehat{\mathrm{C}}_{k_\sigma}(\mathcal{S})$, up to a constant.*

*Proof.* As $U(\mathbf{x}) \to c \, \forall \mathbf{x} \in \mathcal{X}$, the following equality holds:

$$\lim_{U(\mathbf{x}) \to c} \widehat{\mathrm{UC}}_{k_\sigma}(\mathcal{S}) = \lim_{U(\mathbf{x}) \to c} \frac{1}{N} \sum_{n=1}^{N} U(\mathbf{x}_n) \cdot \max_{\mathbf{x}' \in \mathcal{S}} k_\sigma(\mathbf{x}_n, \mathbf{x}')$$

$$= \frac{1}{N} \sum_{n=1}^{N} \lim_{U(\mathbf{x}_n) \to c} U(\mathbf{x}_n) \cdot \max_{\mathbf{x}' \in \mathcal{S}} k_\sigma(\mathbf{x}_n, \mathbf{x}') = c \cdot \widehat{\mathrm{C}}_{k_\sigma}(\mathcal{S}).$$

The last equality holds since each term with limit inside the sum converges to a specific value. □

**Proposition 4.** *Suppose $k_\sigma(\mathbf{x}, \mathbf{x}'; g) = \psi\big((g(\mathbf{x}) - g(\mathbf{x}'))/\sigma\big)$ for a fixed $g : \mathcal{X} \to \mathbb{R}^d$ which is injective on $\mathcal{U}$ and a function $\psi : \mathbb{R}^d \to [0, 1]$ with $\psi(0) = 1$ and for all $t \in \mathbb{R}^d$ with $\|t\| = 1$, $\lim_{a \to \infty} \psi(at) = 0$. If $\sigma \to 0$, the estimated uncertainty coverage $\widehat{\mathrm{UC}}_{k_\sigma}(\mathcal{S})$ approaches $\sum_{s=1}^{|\mathcal{S}|} U(\mathbf{x}_s; f)$, up to a constant.*

*Proof.* As $\sigma \to 0$, the function $k_\sigma$ approaches to the following form:

$$k_\sigma(\mathbf{x}, \mathbf{x}') = \begin{cases} 1 & \text{if } \mathbf{x} = \mathbf{x}', \\ 0 & \text{otherwise.} \end{cases}$$

Then, we have

$$\widehat{\mathrm{UC}}_{k_\sigma}(\mathcal{S}) = \frac{1}{N} \sum_{n=1}^{N} U(\mathbf{x}_n) \cdot \max_{\mathbf{x}' \in \mathcal{S}} k_\sigma(\mathbf{x}, \mathbf{x}')$$

$$= \frac{1}{N} \sum_{n=1}^{N} U(\mathbf{x}_n) \cdot \mathbb{1}\big[\exists \mathbf{x}' \in \mathcal{S} \text{ s.t } \mathbf{x}_n = \mathbf{x}'\big] \propto \sum_{s=1}^{|\mathcal{S}|} U(\mathbf{x}_s)$$

Note that the indicator function is 1 only if $\mathbf{x}_n$ is equal to one of data points in $\mathcal{S}$. □

## A.3 Greedy Algorithm

**Corollary 6.** *In the setting of Theorem 2, let $\hat{\mathcal{S}} \subseteq \mathcal{U}$ be the result of UHerding for $B$ steps to add to $\mathcal{L}$, and $\mathrm{UC}^* = \max_{\mathcal{S} \subseteq \mathcal{U}, |\mathcal{S}|=B} \mathrm{UC}_{k_\sigma}(\mathcal{L} \cup \mathcal{S})$ the optimal coverage obtainable among $\mathcal{U}$. Then*

$$\mathrm{UC}_{k_\sigma}(\mathcal{L} \cup \hat{\mathcal{S}}) \geq \left(1 - \frac{1}{e}\right) \mathrm{UC}^* - \left(2 - \frac{1}{e}\right) U_{\max} \sqrt{\frac{B}{N}} \left[8L_\sigma + \frac{1}{2}\sqrt{d \log\left(R^2 \frac{N}{B}\right) + \frac{2}{B} \log \frac{2}{\delta}}\right].$$

*Proof.* First, $\widehat{\mathrm{UC}}_{k_\sigma}$ is submodular by Lemma 10. Thus, if we let $\bar{\mathcal{S}} \in \arg\max_{\mathcal{S} \subseteq \mathcal{U}: |\mathcal{S}|=B} \widehat{\mathrm{UC}}_{k_\sigma}(\mathcal{S})$, the classical result of Nemhauser et al. (1978) implies that $\widehat{\mathrm{UC}}_{k_\sigma}(\hat{\mathcal{S}}) \geq \left(1 - \frac{1}{e}\right) \widehat{\mathrm{UC}}_{k_\sigma}(\bar{\mathcal{S}})$.

Let $\mathcal{S}^* \in \arg\max_{\mathcal{S} \subseteq \mathcal{U}: |\mathcal{S}|=B} \mathrm{UC}_{k_\sigma}(\mathcal{S})$ be the optimal size-$B$ subset of $\mathcal{U}$ for $\mathrm{UC}_{k_\sigma}$. By definition, $\widehat{\mathrm{UC}}_{k_\sigma}(\bar{\mathcal{S}}) \geq \widehat{\mathrm{UC}}_{k_\sigma}(\mathcal{S}^*)$. Thus, calling the bound on the worst-case absolute error of the coverage estimate $\varepsilon$, it holds with probability at least $1 - \delta$ that

$$\mathrm{UC}_{k_\sigma}(\hat{\mathcal{S}}) \geq \widehat{\mathrm{UC}}_{k_\sigma}(\hat{\mathcal{S}}) - \varepsilon \geq \left(1 - \frac{1}{e}\right) \widehat{\mathrm{UC}}_{k_\sigma}(\mathcal{S}^*) - \varepsilon \geq \left(1 - \frac{1}{e}\right) \mathrm{UC}_{k_\sigma}(\mathcal{S}^*) - \left(2 - \frac{1}{e}\right) \varepsilon. \; \square$$

The following result guarantees that greedy optimization of $\mathrm{UC}_{k_\sigma}$ or $\widehat{\mathrm{UC}}_{k_\sigma}$ achieves a $(1 - 1/e)$ approximation, by Nemhauser et al. (1978).

**Lemma 10.** *For any fixed $\mathcal{L}$, $\mathcal{S} \mapsto \mathrm{UC}_{k_\sigma}(\mathcal{L} \cup \mathcal{S})$ and $\mathcal{S} \mapsto \widehat{\mathrm{UC}}_{k_\sigma}(\mathcal{L} \cup \mathcal{S})$ are nonnegative, submodular, monotone functions.*

*Proof.* We prove that UCoverage is non-negative monotone submodular for any set that contains the fixed set $\mathcal{L}$; this implies that $\widehat{\mathrm{UC}}_{k_\sigma}$ is as well, as it is an instance of $\mathrm{UC}_{k_\sigma}$ with an empirical distribution for $\mathbf{x}$.

For $\forall \mathbf{x}, \mathbf{x}' \in \mathcal{X}$, we assumed that $U(\mathbf{x}) \geq 0$ and $k_\sigma(\mathbf{x}, \mathbf{x}') \geq 0$. Thus, for any subset $\mathcal{A} \subset \mathcal{X}$, $\mathrm{UC}_{k_\sigma}(\mathcal{A}) \geq 0$, which implies $\mathrm{UC}_{k_\sigma}(\mathcal{L} \cup \mathcal{S}) \geq 0$.

Next, we show monotonocity: $\forall \mathcal{L} \subset \mathcal{A} \subset \mathcal{B} \subset \mathcal{X}$, $\mathrm{UC}_{k_\sigma}(\mathcal{A}) \leq \mathrm{UC}_{k_\sigma}(\mathcal{B})$.

$$\mathrm{UC}_{k_\sigma}(\mathcal{B}) = \mathbb{E}_\mathbf{x}[U(\mathbf{x}) \cdot \max_{\mathbf{x}' \in \mathcal{B}} k_\sigma(\mathbf{x}, \mathbf{x}')]$$

$$= \mathbb{E}_\mathbf{x}\left[ U(\mathbf{x}) \cdot \max\left( \max_{\mathbf{x}' \in \mathcal{B} \setminus \mathcal{A}} k_\sigma(\mathbf{x}, \mathbf{x}'), \max_{\mathbf{x}' \in \mathcal{A}} k_\sigma(\mathbf{x}, \mathbf{x}') \right) \right]$$

$$\geq \mathbb{E}_\mathbf{x}[U(\mathbf{x}) \cdot \max_{\mathbf{x}' \in \mathcal{A}} k_\sigma(\mathbf{x}, \mathbf{x}')].$$

Lastly, we show submodularity: $\mathrm{UC}_{k_\sigma}(\mathcal{A} \cup \{\tilde{\mathbf{x}}\}) - \mathrm{UC}_{k_\sigma}(\mathcal{A}) \geq \mathrm{UC}_{k_\sigma}(\mathcal{B} \cup \{\tilde{\mathbf{x}}\}) - \mathrm{UC}_{k_\sigma}(\mathcal{B})$ for $\forall \mathcal{L} \subset \mathcal{A} \subset \mathcal{B} \subset \mathcal{X}$. Using monotonocity we showed, we prove submodularity as:

$$\mathrm{UC}_{k_\sigma}(\mathcal{A} \cup \{\tilde{\mathbf{x}}\}) - \mathrm{UC}_{k_\sigma}(\mathcal{A}) = \mathbb{E}_\mathbf{x}\left[ U(\mathbf{x}) \cdot \max\left( k_\sigma(\mathbf{x}, \tilde{\mathbf{x}}) - \max_{\mathbf{x}' \in \mathcal{A}} k_\sigma(\mathbf{x}, \mathbf{x}'), 0 \right) \right]$$

$$\geq \mathbb{E}_\mathbf{x}\left[ U(\mathbf{x}) \cdot \max\left( k_\sigma(\mathbf{x}, \tilde{\mathbf{x}}) - \max_{\mathbf{x}' \in \mathcal{B}} k_\sigma(\mathbf{x}, \mathbf{x}'), 0 \right) \right]$$

$$= \mathrm{UC}_{k_\sigma}(\mathcal{B} \cup \{\tilde{\mathbf{x}}\}) - \mathrm{UC}_{k_\sigma}(\mathcal{B}). \qquad \square$$

### A.4 CONNECTIONS TO HYBRID METHODS

**Proposition 7** (Weighted $k$-means of Zhdanov 2019). *Define an uncertainty measure $U(\mathbf{x}; f)$ from another uncertainty measure $U(\mathbf{x}; f)$ as $U(\mathbf{x}; f) := U'(\mathbf{x}; f) \cdot \mathbb{1}[U'(\tilde{\mathbf{x}}; f) \geq \nu]$, where $\nu \geq 0$ satisfies $\sum_{n=1}^N \mathbb{1}[U'(\mathbf{x}_n; f) \geq \nu] = M$, a pre-defined number. Then weighted $k$-means with uncertainty $U'$, changed to use greedy kernel $k$-medoids, is UHerding with uncertainty $U$ and the same kernel.*

*Proof.* With the modification of the $k$-means objective into a greedy kernel $k$-medoids objective, the objective of weighted $k$-means with $U'(\mathbf{x})$ as weights can be converted into:

$$\mathbf{x}^* \in \underset{\tilde{\mathbf{x}} \in \mathcal{X}}{\arg\max} \, \mathbb{1}[U'(\tilde{\mathbf{x}}) \geq \nu] \cdot \frac{1}{N} \sum_{n=1}^N U'(\mathbf{x}_n) \cdot \min_{x' \in \mathcal{L} \cup \{\tilde{\mathbf{x}}\}} \|\phi(\mathbf{x}_n) - \phi(\mathbf{x}')\|_{\mathcal{H}}^2 \qquad (5)$$

$$= \underset{\tilde{\mathbf{x}} \in \mathcal{X}}{\arg\max} \, \frac{1}{N} \sum_{n=1}^N U(\mathbf{x}_n) \cdot \min_{x' \in \mathcal{L} \cup \{\tilde{\mathbf{x}}\}} \|\phi(\mathbf{x}_n) - \phi(\mathbf{x}')\|_{\mathcal{H}}^2 \qquad (6)$$

$$= \underset{\tilde{\mathbf{x}} \in \mathcal{X}}{\arg\max} \, \frac{1}{N} \sum_{n=1}^N U(\mathbf{x}_n) \cdot \max_{x' \in \mathcal{L} \cup \{\tilde{\mathbf{x}}\}} k_\sigma(\mathbf{x}_n, \mathbf{x}'). \qquad (7)$$

This is equivalent to the objective of UHerding with $U(\mathbf{x})$ as the choice of uncertainty. $\qquad \square$

**Proposition 8** (ALFA-Mix of Parvaneh et al. 2022). *Let $\hat{y}(\cdot; f)$ be the predicted label of an input under $f$. Define an uncertainty measure*

$$U(\mathbf{x}; f) := \mathbb{1}\left[ \exists \text{ class } j \text{ s.t. } \hat{y}\big(\alpha_j(\mathbf{x})\, g(\mathbf{x}) + (1 - \alpha_j(\mathbf{x}))\, \bar{g}_j; f\big) \neq \hat{y}(g(\mathbf{x}); f) \right] \qquad (3)$$

*where $\bar{g}^j$ is the mean of feature representations belonging to class $j$ and $\alpha_j(\mathbf{x}) \in [0, 1)$ is the same parameter as determined by ALFA-Mix. Then ALFA-Mix, with clustering replaced by greedy kernel $k$-medoids, is UHerding with uncertainty $U$ and the same kernel.*

*Proof.* ALFA-Mix selects closest data points to the center of $k$-means clusters where $k$-means is fitted with filtered data points. It keeps a data point $\mathbf{x}$ if it satisfies that $\exists$ class $j$ s.t. $\hat{y}\big(\alpha_j(\mathbf{x})\, g(\mathbf{x}) + (1 - \alpha_j(\mathbf{x}))\, \bar{g}_j; f\big) \neq \hat{y}(g(\mathbf{x}); f)$, which we can express as the indicator function in Equation (3).

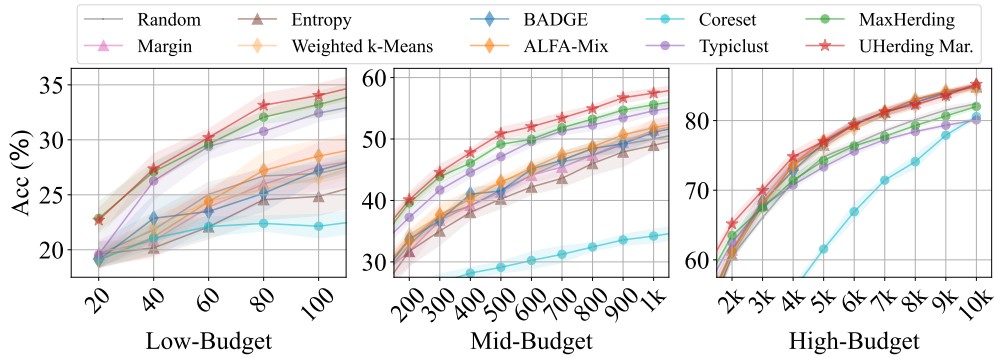

Figure 7: Comparison on CIFAR10 for supervised-learning tasks.

With the replacement of $k$-means with a greedy kernel $k$-medoids objective, the objective of ALFA-Mix can be converted into:

$$\mathbf{x}^* \in \arg\max_{\tilde{\mathbf{x}} \in \mathcal{X}} \frac{1}{N} \sum_{n=1}^{N} U(\mathbf{x}; f) \cdot \max_{\mathbf{x}' \in \mathcal{L} \cup \{\tilde{\mathbf{x}}\}} k_\sigma(\mathbf{x}_n, \mathbf{x}'). \qquad \square$$

**Proposition 9** (BADGE, Ash et al., 2020). *If $\forall \mathbf{x} \in \mathcal{U}$, $\hat{p}(\mathbf{x}; f) \to \frac{1}{K}\vec{1}$, then this BADGE approaches a slightly modified MaxHerding: $\left(\mathbb{1}\left[\hat{y}(\mathbf{x}_n; f) = \hat{y}(\mathbf{x}'; f)\right] - \frac{1}{K}\right) k_\sigma(\mathbf{x}_n, \mathbf{x}'; g)$ instead of $k_\sigma(\mathbf{x}_n, \mathbf{x}'; g)$. If $\sigma \to 0$, it approaches to the uncertainty-based method where uncertainty is defined as, $U''(\tilde{\mathbf{x}}) := \min_{\mathbf{x}' \in \mathcal{L} \cup \{\tilde{\mathbf{x}}\}} \|\hat{y}(\mathbf{x}'; f) - \hat{p}(\mathbf{x}'; f)\|_2^2$.*

*Proof.* Recall $q(\mathbf{x}) = \hat{y}(\mathbf{x}; f) - \hat{p}(\mathbf{x}; f)$. As $\forall \mathbf{x} \in \mathcal{U}$, $\hat{p}(\mathbf{x}; f) \to \frac{1}{K}\vec{1}$, it is true that $\|q(\mathbf{x})\|_2^2 \to 1 - \frac{1}{K}$ and $\langle q(\mathbf{x}), q(\mathbf{x}') \rangle \to \mathbb{1}[\hat{y}(\mathbf{x}; f) = \hat{y}(\mathbf{x}'; f)] - \frac{1}{K}$. With the assumption that $\forall \mathbf{x} \in \mathcal{X}$, $k_\sigma(\mathbf{x}, \mathbf{x}; g) = c$,

$$h(\mathbf{x}_n, \mathbf{x}') \to 2 \left( \mathbb{1}[\hat{y}(\mathbf{x}_n; f) = \hat{y}(\mathbf{x}'; f)] - \frac{1}{K} \right) k_\sigma(\mathbf{x}_n, \mathbf{x}'; g) - 2c \left( 1 - \frac{1}{K} \right). \qquad (8)$$

Therefore, the following is true:

$$\mathbf{x}^* \in \arg\max_{\tilde{\mathbf{x}} \in \mathcal{U}} \frac{1}{N} \sum_{n=1}^{N} \max_{\mathbf{x}' \in \mathcal{L} \cup \{\tilde{\mathbf{x}}\}} h(\mathbf{x}_n, \mathbf{x}) \qquad (9)$$

$$= \arg\max_{\tilde{\mathbf{x}} \in \mathcal{U}} \frac{1}{N} \sum_{n=1}^{N} \max_{\mathbf{x}' \in \mathcal{L} \cup \{\tilde{\mathbf{x}}\}} \left( \mathbb{1}[\hat{y}(\mathbf{x}_n; f) = \hat{y}(\mathbf{x}'; f)] - \frac{1}{K} \right) k_\sigma(\mathbf{x}_n, \mathbf{x}'; g). \qquad (10)$$

If $\sigma \to 0$, $k_\sigma(\mathbf{x}, \mathbf{x}'; g) = \mathbb{1}[\mathbf{x} = \mathbf{x}']$. Then, $\max_{\mathbf{x}' \in \mathcal{L} \cup \{\tilde{\mathbf{x}}\}} h(\mathbf{x}_n, \mathbf{x}') \to \min_{\mathbf{x}' \in \mathcal{L} \cup \{\tilde{\mathbf{x}}\}} \|q(\mathbf{x}_n)\|_2^2 + \|q(\mathbf{x}')\|_2^2 = \min_{\mathbf{x}' \in \mathcal{L} \cup \{\tilde{\mathbf{x}}\}} \|\hat{y}(\mathbf{x}'; f) - \hat{p}(\mathbf{x}'; f)\|_2^2$. Then,

$$\mathbf{x}^* \in \arg\max_{\tilde{\mathbf{x}} \in \mathcal{U}} \frac{1}{N} \sum_{n=1}^{N} \max_{\mathbf{x}' \in \mathcal{L} \cup \{\tilde{\mathbf{x}}\}} h(\mathbf{x}_n, \mathbf{x}) \qquad (11)$$

$$= \arg\max_{\tilde{\mathbf{x}} \in \mathcal{U}} \min_{\mathbf{x}' \in \mathcal{L} \cup \{\tilde{\mathbf{x}}\}} \|\hat{y}(\mathbf{x}'; f) - \hat{p}(\mathbf{x}'; f)\|_2^2 \qquad (12)$$

Therefore, BADGE approaches to $\arg\max_{\tilde{\mathbf{x}} \in \mathcal{U}} U''(\tilde{\mathbf{x}})$ as $\sigma \to 0$. $\qquad \square$

## B  ADDITIONAL COMPARISON WITH STATE OF THE ART

In addition to Figure 4 where we compare state-of-the-art active learning methods on CIFAR100 and TinyImageNet datasets for supervised learning tasks, we provide Figure 7 for additional results on CIFAR10 dataset. Again, we employ a ResNet18 randomly initialized at each iteration.

Similar to the results on CIFAR100 and TinyImageNet datasets, UHerding significantly outperforms representation-based methods in high-budget regimes, and uncertainty-based methods in low- and mid-budget regimes, confirming the robustness of UHerding over other active learning methods.

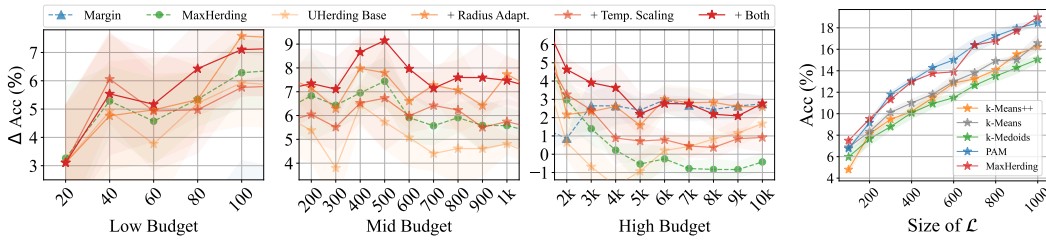

(a) Comparison with and without components of parameter adaptation.

(b) Clustering methods

Figure 8: Comparison of some components in UHerding (Left) and clustering methods (Right)

## C  COMPONENT ANALYSIS OF UHERDING

In this ablation study, we examine the contribution of each component of UHerding to its overall generalization performance. Specifically, we incrementally add each of temperature scaling (Temp. Scaling) and adaptive radius (Radius Adap.) individually to the UHerding baseline, which selects data points based solely on the UHerding acquisition defined in Equation (2). We also evaluate the combined effect of both components. As in Section 4.1, we use CIFAR-10 dataset, and present the results using $\Delta$Acc relative to Random selection to clearly highlight the differences.

Although temperature scaling enhances performance over the UHerding baseline, its performance is still comparable to MaxHerding in the low-budget regime, and worse than Margin in the high-budget regime. Radius adaptation generally matches or exceeds MaxHerding across budget regimes but quickly converges to Margin's performance in the high-budget regime. When both temperature scaling and radius adaptation are applied, performance surpasses both MaxHerding and Margin across all budget regimes, with saturation in the high-budget regime occurring significantly more gradually than with radius adaptation alone. Please note that Margin's performance in the low- and mid-budget regimes is not visible due to its exceptionally poor results relative to other methods.

## D  COMPARISON OF CLUSTERING METHODS

As noted in Section 2, we compare several clustering methods applied to BADGE Ash et al. (2020) to justify the replacement of $k$-means and $k$-means++ of existing active learning methods with Max-Herding. Figure 8b compares $k$-means, $k$-means++, $k$-medoids (iterative optimization), partition around medoids (PAM), and MaxHerding, a greedy kernel $k$-medoids. We train a ResNet18 randomly initialized at each iteration on CIFAR100.

Surprisingly, $k$-medoids performs slightly worse than $k$-means, showing that searching for medoids is not necessarily better than search of means. However, the gap between PAM and $k$-medoids shows that optimization methods do make significant changes. Although PAM works the best overall, MaxHerding is comparable with much less computation Bae et al. (2024). It justifies the replacement of $k$-means and $k$-means++ with MaxHerding for existing active learning methods.

