# OpenReview forum: "Uncertainty Herding: One Active Learning Method for All Label Budgets"
_ICLR.cc/2025/Conference — ICLR 2025 Poster_

### Official Review · Reviewer_Bjve · 2024-11-02

**Soundness:** 2
**Presentation:** 2
**Contribution:** 2
**Rating:** 6
**Confidence:** 4

**Summary:**

This paper interpolates between the low-budget and the high-budget regime active learning (AL) by an objective called uncertainty coverage. The proposed algorithm Uncertainty Herding is a combination of the low-budget AL algorithm MaxHerding (Bae et al. 2024) and the classical uncertainty sampling in a way that the objective in MaxHerding is re-weighted by an uncertainty function. There are three improvements other than the objective: the temperature scaling (to weaken the effects of uncertainty functions when low-budget), the radius adaptation (to narrow down the coverage when high-budget), and the entropy regularization. The authors conduct experiments showing the superiority of the algorithm.

**Strengths:**

1. The proposed algorithm is a natural interpolation between the low-budget MaxHerding and the high-budget uncertainty sampling. The authors provide a smart way (temperature scaling and radius adaptation) to decide each algorithm's proportion and also prove the equivalence in an asymptotic sense.
2. The algorithm has reached a performance surpassing the existing benchmarks, especially the other attempts to interpolate between the low-budget and the high-budget regimes.
3. The algorithm's computational burden is acceptable.

**Weaknesses:**

1. (Major) The implemented algorithm in the main experiments consists of three improvements compared to the naive Uncertainty Herding: the temperature scaling, the radius adaptation, and the entropy regularization. From my point of view, this paper's main contribution is giving a natural (and probably SOTA) interpolation between the low-budget and the high-budget AL regimes. In that sense, the first two terms seem very natural, yet the third term is unnatural and not analyzed in the paper.  On the one hand, the third term is important (and sometimes even critical) since the third term does boost the performance of the proposed algorithm. For example, in Appendix C, without entropy-regularization, the performance of Uncertainty Herding does not exceed uncertainty sampling in the high-budget regime. Intuitively, without that regularization, it would be weird that the algorithm (as an interpolation of the MaxHerding and the margin-based uncertainty sampling) outperforms both the MaxHerding and the margin-based uncertainty sampling in EVERY regime. However, the ablation study only considers adding entropy-regularization to the random sampling (passive learning). The authors make no attempts to explain or identify the effects of the entropy-regularization, which makes the logic incomplete.
2. (Moderate) The theory part does not contain much contribution and novelty.
3. (Moderate) There seem to be no descriptions of how the temperature scaling changes the uncertainty function. For example, the $f_{\tau}$ function in Definition 5 is used without definition.
4. (Minor) Figures in the main text are too small.

**Questions:**

1. The entropy-regularization: Is it just an engineering trick or a vital part of active learning? Would you mind conducting some further ablation studies, for example, combining entropy-regularization with other active learning algorithms (the MaxHerding and the margin-based uncertainty sampling)?
2. The framework: This paper unifies other hybrid methods in the uncertainty coverage objective by choosing different uncertainty functions and kernels. Does this framework provide any (theoretical or empirical) insights into selecting the uncertainty functions or the kernels in practice?

---

> ### Author Response · Authors · 2024-11-21
> **Response, part 1**
>
> Thank you for your feedback and your work in reviewing our paper.
>
>
> ## Weaknesses
> > (Major) …In that sense, the first two terms seem very natural, yet the third term is unnatural and not analyzed in the paper. On the one hand, the third term is important (and sometimes even critical) since the third term does boost the performance of the proposed algorithm. For example, in Appendix C, without entropy-regularization, the performance of Uncertainty Herding does not exceed uncertainty sampling in the high-budget regime…
>
> We very much appreciate this comment, which made us think more deeply about the inclusion of entropy regularization. In developing the method, we added it as another way to try to assist calibration, but on reflection and further ablation studies, we think that the extra complexity is just not worth the fairly minor performance improvement it brings. We have decided to completely remove entropy regularization from our paper, and updated all the relevant text and figures with results that no longer include it. The changes are summarized as follows:
>
> 1. We re-ran all the proposed UHerding method without entropy regularization and updated the performance of UHerding in Figure 1-8 as well as Table 1.
> 2. We removed the paragraph explaining entropy regularization under “Handling the low-budget case: calibration” of Section 3.2.
> 3. We also removed an ablation study for entropy regularization previously listed under Appendix D.
>
> All the changes are colored in blue in the revised manuscript. Without entropy regularization, the performance is sometimes worse, but sometimes better, compared to previous results with entropy regularization. In the cases where performance is worse, the gap is not large enough for any competitor to ever outperform
> UHerding, and for each competitor there remains some budget where UHerding wins substantially. It remains that UHerding significantly outperforms uncertainty-based methods in low- and mid-budget regimes whereas it outperforms representation-based methods in high-budget regimes.
>
> For overall quantitative results, please refer to the updated Table 1, where UHerding is still the best (colored in green) compared to other methods on average throughout all budget regimes.
>
> Again, we greatly appreciate this push to make our method more concise and straightforward.
>
>
>
> > (Moderate) The theory part does not contain much contribution and novelty.
>
> Prior works on low-budget active learning, including MaxHerding (Bae et al., 2024) and ProbCover (Yehuda et al., 2022), provide lower bounds for $\hat{M}(\hat{S})$ in terms of $\hat{M}(S^\*)$, where $\hat{M}$ represents the approximate (empirical) value of the measure, e.g. GCoverage; $\hat{S}$ is the greedily selected set; and $S^\*$ is the set which optimizes $\hat{M}$ under a budget constraint. Our theoretical contribution seeks to go further. Specifically, we aim to assess the quality of greedy selection in terms of the true measure $M$, rather than its approximation $\hat{M}$.
>
> To address this, we introduced Theorem 2 and Corollary 6, which establish a lower bound for $M(\hat{S})$ in terms of $M(S^*)$. As far as we are aware, this perspective – analyzing bounds with respect to the true measure $M$ – has not been addressed in previous active learning research.
>
> These statements help justify the importance of a sufficiently large unlabeled set $\mathcal{U}$ relative to the acquisition budget $B$. Moreover, our analysis provides an explanation for the sensitivity of ProbCover to the radius parameter $\sigma$, as noted through the Lipschitz property of the function $k_\sigma$ (see line 208).
>
> In terms of novelty: it is the case that our proof techniques are standard in learning theory, and we do not claim otherwise. Even so, we think the contribution is novel. Most low-budget active learning papers did not even attempt to address this problem; others, such as Bae et al. (2024), made simple claims based on a Hoeffding inequality that ignore the key point that the set $\\mathcal S$ is *not independent* of the points being used to estimate $\\hat M$.
>
> If there are further specific aspects where our theoretical contributions could be clarified or strengthened, we would appreciate additional guidance. We are eager to improve our work and welcome any insights.
>
>
>
> > (Moderate) There seem to be no descriptions of how the temperature scaling changes the uncertainty function. For example, the $f\_\tau$ function in Definition 5 is used without definition.
>
> We initially assumed that temperature scaling, as a widely recognized post-hoc calibration technique (Guo et al., 2017; Platt, 1999), might not need further explanation than the citation and its brief definition in line 248 of the original submission. However, we agree that additional clarification could be helpful, especially for readers who may be less familiar with model calibration. We have thus expanded the description slightly in lines 249-251.

---

> > ### Comment · Reviewer_Bjve · 2024-11-23
> >
> > I'm glad that the authors appreciate my advice and thanks for the efforts. Now my major concern is fully addressed, so I will adjust my rating (from 5 to 6).

---

> > > ### Author Response · Authors · 2024-11-23
> > > **Response to Reviewer Bjve**
> > >
> > > We sincerely appreciate the reviewer for taking the time to re-evaluate our work and adjust the score accordingly. The feedback provided has been invaluable in helping us further improve our work. Thank you.

---

> ### Author Response · Authors · 2024-11-21
> **Response, part 2**
>
> > (Minor) Figures in the main text are too small.
>
> We appreciate this suggestion. We have reviewed all figures and increased the font size, especially in Figures 1 and 2, to ensure readability. If any figures remain difficult to read, please let us know and we will address it further.
>
>
> ## Questions
>
>
> > The entropy-regularization: Is it just an engineering trick or a vital part of active learning? Would you mind conducting some further ablation studies, for example, combining entropy-regularization with other active learning algorithms (the MaxHerding and the margin-based uncertainty sampling)?
>
> As discussed above, we have completely removed entropy regularization from our paper, since your questioning prompted us to recognize that it indeed was not a vital component.
>
>
> > The framework: This paper unifies other hybrid methods in the uncertainty coverage objective by choosing different uncertainty functions and kernels. Does this framework provide any (theoretical or empirical) insights into selecting the uncertainty functions or the kernels in practice?
>
> We appreciate this question, and will add some more discussion of it to the final paper.
>
> From our experiments, it appears that the specific choice of uncertainty function $U(\mathbf{x}; f\_\tau)$ (e.g., Entropy, Margin, or Confidence) or the precise form of a radial basis-type kernel function $k\_\sigma(\mathbf{x}, \mathbf{x}')$ (e.g., Gaussian, Student-t, or Laplace) is not crucial to make active learning algorithms robust to different budget regimes.
>
> In particular, Figure 3 shows that, while Margin performs slightly better than Entropy and Confidence, the differences between these uncertainty functions (blue lines) are not substantial. A similar trend is observed for UHerding, where different uncertainty measures yield comparable performance (red lines).
>
> For kernel choice, results from Figure 7 in MaxHerding (Bae et al., 2024) suggest that, among common kernels like Gaussian, Student-t, Laplace, and Cauchy, the impact on generalization performance is minimal in active learning tasks. As discussed by Bae et al. (2024), however, the top-hat function implicitly used by ProbCover does seem to be a poor choice; this is consistent with our analysis in Theorem 2, which requires a Lipschitz kernel to show good generalization, which the top-hat function is not (but the others are).
>
> Our findings indicate that the relative weighting between uncertainty and kernel functions across budget regimes is far more impactful. Accordingly, we provide both theoretical and empirical methods for smoothly interpolating between low- and high-budget regimes, by adjusting parameters such as $\tau$ in $U(\mathbf{x}; f\_\tau)$ and $\sigma$ in $k\_\sigma(\mathbf{x}, \mathbf{x}')$, making active learning more adaptive across budget regimes.
>
> While reformulating existing hybrid methods, particularly BADGE, as UHerding does offer performance improvements (see Figure 6a), parameter adaptation seems more important. Thus, the main insight is to rely more on good representations in low-budget regimes, and increasingly leverage uncertainty computed using a classifier as more data points become available. Given these considerations, selecting any reasonable uncertainty and kernel functions should suffice.

---

### Official Review · Reviewer_DKgc · 2024-11-04

**Soundness:** 3
**Presentation:** 3
**Contribution:** 3
**Rating:** 8
**Confidence:** 3

**Summary:**

This paper proposes an active learning method suitable for both low- and high-budget regimes. It defines an "uncertainty coverage" metric, estimated through its empirical counterpart. An uncertainty herding method is then introduced, based on the greedy optimization of this estimated uncertainty coverage objective. Numerical experiments are conducted within a transfer learning application.

**Strengths:**

- This paper is largely well-written, with extensive discussions on the existing literature.
- The numerical experiments are well-executed.

**Weaknesses:**

- The proposed method seems to rely more on a combination of heuristics and lacks guiding principles. Some analysis is also presented only in the asymptotic regime. In particular, Proposition 4 requires $\sigma\rightarrow 0$.
- A more intriguing question is that the paper does not provide a detailed recipe for choosing the design or detecting whether the budget is high or low.
- In terms of writing, although this paper builds on a line of literature, the exposition is far from self-contained; some notation and basic definitions are absent from Section 3.1.
- Section 3.4 is poorly organized. Instead of simply citing lengthy theorems, it would provide more insights by including a coherent analysis and direct comparisons.
- Figures 3 and 4: The plots use the same markers for different methods.

**Questions:**

See weakness.

---

> ### Author Response · Authors · 2024-11-21
> **Response, part 1**
>
> Thank you for your feedback and your work in reviewing our paper.
>
> ## Weaknesses
>
> > The proposed method seems to rely more on a combination of heuristics and lacks guiding principles. Some analysis is also presented only in the asymptotic regime. In particular, Proposition 4 requires $\sigma \rightarrow 0$.
>
> We acknowledge that our method incorporates some heuristic elements, particularly for the choice of adaptive parameters. However, we would like to emphasize that these heuristics are grounded in theoretical analysis, as described in Propositions 3 and 4.
>
> These propositions theoretically demonstrate how Uncertainty Coverage (UCoverage) can approximate Generalized Coverage (GCoverage) in low-budget regimes, and align with uncertainty measures in high-budget regimes. This analysis is based on the observations that representation-based methods work well in low-budget regimes, whereas uncertainty-based methods perform well in high-budget regimes. We believe this theoretical motivation is not only a core guiding principle for our UHerding method but also highly relevant for future research in active learning.
>
> We would also like to highlight that the primary quantitative analysis, particularly Theorem 2 and Corollary 6, are finite-sample bounds that hold for any parameter values or dataset size.
>
>
> > A more intriguing question is that the paper does not provide a detailed recipe for choosing the design or detecting whether the budget is high or low.
>
> We would like to clarify that the main goal of our paper is to avoid the need to decide whether the budget is high or low. We think this notion of discrete budget regimes, while useful, does not reflect the reality of active learning tasks (as discussed in Section 1). It seems like budget regimes are typically shaped by multiple continuous factors, including data distribution characteristics, the number of labeled data points, and model capacity, making it challenging to establish a single, clear boundary.
>
> Even if we can find a reasonable boundary, it is more intuitive and natural to have continuous transition between different budget regimes, rather than abruptly shifting from low to high-budget at a certain point. As a result, we proposed a method that smoothly interpolates between low- and high-budget regimes. While we continue to refer to “low”, “mid”, and “high” budget regimes to give a general sense of different budget levels in our experimental analysis, the proposed UHerding method is not aware of these labels in any way.
>
> We are unsure whether by "a detailed recipe for choosing the design" you meant the "design" of budget levels (just discussed), or something else. While our framework is flexible, we emphasize that we do suggest a default form of the kernel and a hyperparameter-free method to adapt the temperature and radius parameters, which is common to all of our experiments.
>
>
> > In terms of writing, although this paper builds on a line of literature, the exposition is far from self-contained; some notation and basic definitions are absent from Section 3.1.
>
> We appreciate this concern. We have introduced the main notations and basic definitions in Section 2: Related Work and Background, which, to our understanding, is standard practice when a background section is included. Given the limited space, reintroducing these elements in Section 3 would reduce efficiency.
>
> That said, we identified some instances, such as $\mathcal{X}$ and $k_\sigma$, where notations were not adequately clarified, and we have now added these definitions to enhance readability in the revised manuscript. If further clarification is needed in any place, please let us know; we would be very happy to improve the clarity of our paper.

---

> ### Author Response · Authors · 2024-11-21
> **Response, part 2**
>
> > Section 3.4 is poorly organized. Instead of simply citing lengthy theorems, it would provide more insights by including a coherent analysis and direct comparisons.
>
> We appreciate this feedback as well. The propositions in Section 3.4 are intended to clarify the equivalence between UHerding and existing methods, such as weighted $k$-means and ALFA-Mix, under specific conditions. Given space constraints, while we originally wrote longer descriptions in this section, we believe that in the current form each proposition is concise (four or five lines of text, hardly "lengthy" in our opinion) but self-contained to ensure clarity.
>
> For example, Proposition 7 shows that the weighted $k$-means algorithm with uncertainty $U'$ is equivalent to UHerding with uncertainty $U$ under a shared kernel, and we provide the necessary condition on $U$ in terms of $U’$ for this equivalence to hold. Similarly, Proposition 8 establishes that ALFA-Mix can be viewed as UHerding with the uncertainty measure $U$ defined in Eq. (3). By specifying the appropriate form of $U$ for each method, we illustrate how UHerding can generalize these approaches with appropriate adjustments to the uncertainty measure.
>
>  If there are specific parts that could be improved for clarity or coherence, we would greatly appreciate further suggestions and look forward to incorporating the reviewer’s constructive feedback.
>
>
>
>
> > Figures 3 and 4: The plots use the same markers for different methods.
>
> Thank you to the reviewer for this helpful suggestion. After experimenting with different options to better differentiate the methods, we decided to use different markers to represent different active learning categories – random, uncertainty-based, representation-based, and hybrid methods – making it easier for readers to identify each approach at a glance. The changes are reflected to the revised manuscript for all the relevant figures, including Figure 3 and 4. If you have suggestions for further improvement of our visualizations, we would be very happy to hear them.

---

> > ### Comment · Reviewer_DKgc · 2024-11-27
> > **Thank you and score adjusted**
> >
> > Thank you for the helpful clarification. I have increased my score accordingly.

---

> > > ### Author Response · Authors · 2024-11-28
> > > **Response to Reviewer DKgc**
> > >
> > > We greatly appreciate the reviewer for reviewing the rebuttal and re-evaluating our work accordingly. The reviewer's insightful suggestions have significantly enhanced the clarity and quality of this work. Thank you.

---

### Official Review · Reviewer_x6W9 · 2024-11-07

**Soundness:** 3
**Presentation:** 3
**Contribution:** 3
**Rating:** 6
**Confidence:** 2

**Summary:**

The authors propose a method to adapt to different budget levels in active learning. They theoretically prove that the proposed method achieves near optimal distribution level coverage. Empirically, they show that their method outperforms state of the art.

**Strengths:**

The theoretical results provide insights into algorithm design, especially how to make proposed method robust across different budget levels.

**Weaknesses:**

in figure 5, there is a sudden increase in difference in accuracy from 7.2k to 8k budget. This increase applies to all methods and goes against the general trend observed in figure 5, 6, and 7. Could the authors provide some insight on why that may happen? A similar hikes happens in figure 6(a).

I suspect this is due to the small amount of random seeds used. The error bars in figure 6(a) are quite high, and these hikes might be caused by an outlier point. Could the authors provide results for at least a subset of experiments with 10 or more random seeds and report the median? That may better illustrate the trend.

**Questions:**

can the authors make the legend in figure 1(a) larger and more readable?

---

> ### Author Response · Authors · 2024-11-21
>
> Thank you for your feedback and your work in reviewing our paper.
>
> ## Weaknesses
>
>
> > In figure 5, there is a sudden increase in difference in accuracy from 7.2k to 8k budget. [...]
>
> We appreciate your flagging of this point. We first want to clarify that the y-axis in both Figures 5 and 6 are $\Delta$ Acc relative to the performance of Random selection. These hikes happen when the performance of Random selection does not increase, whereas other methods improve. For example,  the table shows the absolute performance of each method in Figure 5a with the budget size from 7.2k to 8k. Here, the performance of Random selection decreases by about 0.5% whereas that of the other methods improves by close to 1%, which results in such a hike that the reviewer pointed out. The degradation of performance with more number of labeled set as shown in the case of Random selection is in fact not uncommon in active learning. Although it is more common with warm-start than cold-start (Ash et al., 2020), it still does happen with cold-started active learning; for examples, please refer to Figure 16, 24-34 and more in the appendix of ALFA-Mix (Parvaneh et al., 2022).
>
>
> | CIFAR100                      | 7.2k  | 8k    | $\Delta$ (8k, 7.2k) |
> | ----------------------------- | ----- | ----- | ------------------- |
> | Random                        | 74.59 | 74.08 | **-0.51**             |
> | Margin                        | 75.28 | 76.14 | **0.86**                |
> | $\Delta$ (Margin, Random)     | 0.69  | 2.06  | 1.37                |
> | MaxHerding                    | 73.50 | 74.52 | **1.02**                |
> | $\Delta$ (MaxHerding, Random) | -1.09 | 0.44  | 1.53                |
>
> Sometimes, instead, the performance of Random improves only very marginally, as shown in the subset of Figure 6a. Compared to the improvement of Random between 3k and 4k: 0.07%, the improvement of other methods are significantly higher – ActiveFT: 3.46%, MaxHerding: 2.16% and UHerding: 3.85%.
>
>
> | ImageNet                     | 3k    | 4k    | $\Delta$ (4k, 3k) |
> | ---------------------------- | ----- | ----- | ----------------- |
> | Random                       | 32.48 | 32.55 | **0.07**              |
> | ActiveFT                     | 37.02 | 40.48 | **3.46**              |
> | $\Delta$(ActiveFT, Random)   | 4.54  | 7.93  | 3.39              |
> | MaxHerding                   | 42.13 | 44.29 | **2.16**              |
> | $\Delta$(MaxHerding, Random) | 9.65  | 11.74 | 2.09              |
> | UHerding                     | 42.55 | 46.40 | **3.85**              |
> | $\Delta$(UHerding, Random)   | 10.07 | 13.85 | 3.78              |
>
> As suggested by the reviewer, we further investigated this phenomenon by conducting more experiments on Random selection, since it is caused by degradation / almost no improvement of Random selection. We ran Random selection with 10 seeds on CIFAR100 for Figure 5a and ImageNet for Figure 6a, and obtained the following results. On CIFAR100, the performance of Random selection no longer decreases from 7.2k to 8k and the hike is significantly flattened out, although it still exists — $\Delta$(Margin, Random): $1.37 \rightarrow 0.59$, and $\Delta$(MaxHerding, Random): $1.53 \rightarrow 0.75$.
> For ImageNet, the gap between 3k and 4k for Random selection increases somewhat ($0.07 \rightarrow 0.73$) but the gap with other methods e.g. $\Delta$(ActiveFT, Random) at 4k is still larger than 3k.
>
> | CIFAR100                     | 7.2k  | 8k    | $\Delta$ (8k, 7.2k) |
> | ---------------------------- | ----- | ----- | ------------------- |
> | Random                       | 74.21 | 74.48 | 0.27                |
> | $\Delta$(Margin, Random)     | 1.07  | 1.66  | 0.59                |
> | $\Delta$(MaxHerding, Random) | -0.71 | 0.04  | 0.75                |
>
>
>
> | ImageNet                     | 3k    | 4k    | $\Delta$ (4k, 3k) |
> | ---------------------------- | ----- | ----- | ----------------- |
> | Random                       | 32.58 | 33.31 | 0.73              |
> | $\Delta$(ActiveFT, Random)   | 4.44  | 7.17  | 2.73              |
> | $\Delta$(MaxHerding, Random) | 9.55  | 10.98 | 1.43              |
> | $\Delta$(UHerding, Random)   | 9.97  | 13.09 | 3.12              |
>
> According to our observations, it seems like there is more than large variance of Random selection for these “hikes”. Given that it is persistent in other works as we mentioned earlier, it would be an interesting and important direction to investigate further.
>
>
> ## Questions
>
>
> > Can the authors make the legend in figure 1(a) larger and more readable?
>
> As suggested, we changed Figure 1(a) to be larger and readable along with similar changes to Figure 1(b) in the revised manuscript. Please let us know if there are any issues remaining with the modified figures.

---

> ### Author Response · Authors · 2024-12-02
> **Reminder for the End of Discussion**
>
> Dear Reviewer x6W9,
>
> Thank you once again for your thoughtful feedback, which has been valuable to improve our work. We have provided detailed responses to address all the points you raised.
> As the discussion phase is approaching its end, we would greatly value any additional thoughts or feedback you may have regarding our rebuttal. Your input would be very helpful as we work to improve the clarity and quality of our submission.
>
> Thank you for your time and dedication to this review process.

---

### Official Review · Reviewer_jSMZ · 2024-11-09

**Soundness:** 3
**Presentation:** 4
**Contribution:** 4
**Rating:** 8
**Confidence:** 3

**Summary:**

This paper introduces a novel active learning method called Uncertainty Herding (UHerding), which addresses challenges respect to label budget in active learning regimes. Traditional active learning methods often consider either low- or high-budget contexts but not both. This method bridges this gap by incorporating an uncertainty coverage objective, which generalizes both low- and high-budget objectives. This adaptive method achieves state-of-the-art results across various benchmarks, providing a reliable approach for tasks requiring both representation-based and uncertainty-based selection strategies.

**Strengths:**

UHerding successfully addresses both low- and high-budget scenarios in a unified approach, eliminating the need to switch frameworks and tackling the practical challenge of unclear budget boundaries. This method is also practical, as it doesn’t demand high training costs. This innovative approach is a strong contribution.

The paper provides rigorous theoretical analysis into the estimation quality and parameter adaptation of UHerding. Experiments span multiple datasets and scenarios, showing UHerding’s good performance compared to existing methods across various budget levels.

**Weaknesses:**

The method relies on having suitable pre-trained feature extractors and an accurate approximation of the data distribution, which might not always be feasible in real-world situations.

While it shows strong performance in supervised tasks, its effectiveness in transfer learning could benefit from additional validation against approaches tailored to specific domains.

**Questions:**

1. How sensitive is UHerding to changes in the adaptation parameters, particularly when applied to diverse datasets or models? Would a suboptimal parameter choice significantly affect performance?

2. How well does UHerding perform in low-data or imbalanced data scenarios in practice, where pre-trained features may not fully capture distributional or distinguishable differences?

---

> ### Author Response · Authors · 2024-11-21
> **Response, part 1**
>
> Thank you for your feedback and your work in reviewing our paper.
>
> ## Weaknesses
>
>
> > The method relies on having suitable pre-trained feature extractors and an accurate approximation of the data distribution, which might not always be feasible in real-world situations.
>
> We appreciate this interesting point. In the modern deep learning era, it is very common to use pre-trained feature extractors, also known as foundation models; it is perhaps harder to find tasks where people do _not_ utilize them. Foundation models are often very generalizable even for quite severe domain shift.
>
> Even without the presence of a relevant foundation model, we want to emphasize that in pool-based active learning, we assume that we have a fairly large pool of unlabeled data points from which we select data points to annotate. In many domains, it is easy to find large amounts of unlabeled data. We can then train unsupervised/self-supervised feature extractors on the pool.
>
> Even if there is not enough unlabeled data points or pre-trained feature extractors, we can still learn meaningful features from a limited number of unlabeled data points. In the response below to your final question, we show that only with a small fraction of data points, we can extract reasonably useful features to learn similarity between data points – essential information for kernel-based active learning methods such as MaxHerding and UHerding.
>
> In summary, we argue that it is not hard to find pre-trained feature extractors in many domains in modern deep learning era. Even if they are not available, we can collect unlabeled data and train custom feature extractors using advanced self-supervised learning methods such as SimCLR and DINO. Even with a limited number of unlabeled data points, we can still extract meaningful features to estimate similarity between data points.
>
>
> > While it shows strong performance in supervised tasks, its effectiveness in transfer learning could benefit from additional validation against approaches tailored to specific domains.
>
> We thank the reviewer for this thoughtful suggestion. First, we would like to emphasize that we have already included several methods that report results on transfer learning tasks, demonstrating their effectiveness relative to others at the time of their publication: ALFA-Mix, MaxHerding, and ActiveFT.
>
> Among these, ActiveFT is perhaps most explicitly designed for transfer learning. However, we found that it performs significantly worse than UHerding, MaxHerding, and even ALFA-Mix after a few iterations. We believe this is because ActiveFT only considers single-iteration queries, which differs from the iterative querying approach typical in active learning. For this reason, we are uncertain that methods labeled as “tailored” for transfer learning necessarily provide the most meaningful comparisons.
>
> However, we are eager to continue improving our work. If you have any specific suggestions for active learning methods optimized for transfer learning, we would be glad to explore and benchmark these against our method.

---

> ### Author Response · Authors · 2024-11-21
> **Response, part 2**
>
> ## Questions
>
>
> > How sensitive is UHerding to changes in the adaptation parameters, particularly when applied to diverse datasets or models? Would a suboptimal parameter choice significantly affect performance?
>
> We first would like to emphasize that we do not introduce any hyperparameters to tune; our procedure adapts the temperature $\\tau$ and radius $\\sigma$ itself. However, we understand the reviewer’s concern regarding the sensitivity of these choices. To validate the sensitive of the choice of adaptation parameters, we scaled up and down the selected temperature $\tau^\*$ and radius $\sigma^\*$ with scaling factor of $\pm 10, 20,$ and $40$%, and reported the results for one random seed in the table below. Please note that scale **1.00**  corresponds to the “optimal” selection we proposed.
>
> According to the results on temperature, it does not seem like the test performance (in accuracy) is sensitive to the choice of the temperature parameter. Even when scaling factor is $\pm 40$%, the performance degrades only for about $2$%. Even so, as the size of labeled points increases, it quickly catches up the performance; when it is in the high-budget regime (the labeled set size $\mathcal{L} \geq 1000$), the gap with the value selected by our method does not seem to be substantial.
>
> | Method               | Scale    | 40        | 80        | 100       | 400       | 800       | 1000      | 4000      | 8000      | 10000     |
> | -------------------- | -------- | --------- | --------- | --------- | --------- | --------- | --------- | --------- | --------- | --------- |
> | Temperature $\tau^*$ | 1.40     | 26.91     | 31.92     | 34.99     | 46.83     | 54.06     | 57.27     | 75.65     | 82.88     | 85.32     |
> |                      | 1.20     | 27.36     | 33.67     | 33.52     | 47.61     | 56.66     | 57.24     | 73.79     | 82.63     | 84.20     |
> |                      | 1.10     | 28.72     | 33.63     | 33.21     | 47.77     | 55.13     | 58.27     | 75.26     | 82.25     | 84.91     |
> |                      | **1.00** | **28.85** | **33.12** | **35.57** | **48.92** | **55.42** | **57.78** | **74.25** | **82.17** | **85.05** |
> |                      | 0.90     | 28.57     | 33.21     | 33.75     | 47.60     | 55.11     | 57.93     | 75.11     | 83.16     | 84.32     |
> |                      | 0.80     | 26.36     | 34.34     | 34.81     | 47.41     | 52.96     | 58.10     | 73.55     | 82.30     | 84.77     |
> |                      | 0.60     | 26.54     | 32.85     | 35.17     | 49.23     | 53.13     | 57.86     | 73.26     | 82.96     | 84.82     |
>
> Similarly, the test performance is also not very sensitive to the radius parameter $\sigma^*$. Although there is some degradation of performance, particularly in the mid-budget regime ($100 \leq \mathcal{L} \leq 1000$) for scale factor of $\pm 40$%, the gap is insignificant in other cases and regimes.
>
> | Method            | Scale    | 40        | 80        | 100       | 400       | 800       | 1000      | 4000      | 8000      | 10000     |
> | ----------------- | -------- | --------- | --------- | --------- | --------- | --------- | --------- | --------- | --------- | --------- |
> | Radius $\sigma^*$ | 1.40     | 28.60     | 33.44     | 35.67     | 45.87     | 53.58     | 54.89     | 72.16     | 82.03     | 83.90     |
> |                   | 1.20     | 27.68     | 35.34     | 35.65     | 47.56     | 56.50     | 58.18     | 73.17     | 82.28     | 84.51     |
> |                   | 1.10     | 28.83     | 34.08     | 35.59     | 46.12     | 55.73     | 58.81     | 74.11     | 83.15     | 85.25     |
> |                   | **1.00** | **28.85** | **33.12** | **35.57** | **48.92** | **55.42** | **57.78** | **74.25** | **82.17** | **85.05** |
> |                   | 0.90     | 28.85     | 33.13     | 35.57     | 48.92     | 55.42     | 57.80     | 74.25     | 82.17     | 85.05     |
> |                   | 0.80     | 27.09     | 33.22     | 33.41     | 48.32     | 53.98     | 58.37     | 73.35     | 82.05     | 85.38     |
> |                   | 0.60     | 28.73     | 32.55     | 34.28     | 44.08     | 53.90     | 56.90     | 74.83     | 81.90     | 84.88     |

---

> ### Author Response · Authors · 2024-11-21
> **Response, part 3**
>
> > How well does UHerding perform in low-data or imbalanced data scenarios in practice, where pre-trained features may not fully capture distributional or distinguishable differences?
>
> We first want to emphasize that all the low-budget active learning methods such as Typiclust, ProbCover and MaxHerding rely on a pre-trained feature extractors to measure the distance between data points properly. All the previous works utilize unlabeled data points to learn a good feature extractor using self-supervised learning techniques such as SimCLR or DINO.
>
> Although training a good feature extractor using a limited number of unlabeled data points is out of scope of this work – this is one of the most fundamental problems in many machine learning tasks, including generative modeling like large language models (LLMs) and self-supervised learning tasks related to foundation models – as requested by the reviewer, we have conducted some experiments with a limited number of unlabeled data points, and provided a result in the table below. We trained a ResNet18 feature extractor using SimCLR on 0, 5k, 10k and 25k data points of CIFAR10 out of 50k data points that we initially used. (When training on 0 data points, we simply use the randomly initialized network.)
>
> The results demonstrates that even when we use no unlabeled data points, the performance is comparable to Margin selection, which shows that even a randomly initialized feature extractor provides some useful information through the inductive bias of the network design (convolutions, Lipschitz activations, etc). With even a small number of additional unlabeled data points, e.g. 5k, the performance quickly improves and outperforms Margin selection in the low- and mid-budget regimes (labeled set size $\leq 1000$), and MaxHerding in the high-budget regime.
>
>
> | Method     | Num of Unlabeled | 20    | 60    | 100   | 200   | 400   | 600   | 800   | 1000  | 2000  | 4000  | 6000  | 8000  | 10000 |
> | ---------- | ---------------- | ----- | ----- | ----- | ----- | ----- | ----- | ----- | ----- | ----- | ----- | ----- | ----- | ----- |
> | Random     | —                | 19.59 | 25.04 | 26.93 | 32.77 | 39.12 | 44.05 | 47.33 | 50.01 | 60.54 | 71.20 | 76.62 | 80.12 | 82.45 |
> | Margin     | —                | 19.68 | 23.88 | 27.57 | 31.70 | 39.21 | 44.11 | 47.39 | 51.43 | 61.41 | 73.84 | 79.59 | 82.56 | 85.20 |
> | MaxHerding | —                | 22.86 | 29.61 | 33.22 | 39.60 | 46.07 | 49.96 | 53.24 | 55.58 | 63.51 | 71.42 | 76.36 | 79.29 | 82.02 |
> | UHerding   | 0                | 18.72 | 25.13 | 26.36 | 32.12 | 37.63 | 43.13 | 47.35 | 49.54 | 61.07 | 73.95 | 79.99 | 82.71 | 84.83 |
> |            | 5k               | 21.37 | 27.57 | 31.34 | 37.43 | 44.65 | 48.88 | 52.02 | 55.22 | 63.57 | 73.65 | 79.84 | 82.14 | 84.97 |
> |            | 10k              | 21.71 | 28.52 | 32.12 | 38.61 | 46.73 | 50.60 | 54.35 | 56.86 | 64.77 | 73.75 | 79.60 | 82.28 | 84.82 |
> |            | 25k              | 22.07 | 28.93 | 33.64 | 39.14 | 47.36 | 51.61 | 53.90 | 55.72 | 65.36 | 73.51 | 79.11 | 82.65 | 85.04 |
> |            | 50k              | 22.70 | 30.21 | 34.03 | 40.11 | 47.78 | 52.01 | 54.93 | 57.48 | 65.17 | 74.83 | 79.39 | 82.31 | 85.22 |

---

> ### Author Response · Authors · 2024-12-02
> **Reminder for the End of Discussion**
>
> Dear Reviewer jSMZ,
>
> Thank you for providing constructive feedback on our submission. We have carefully addressed all the concerns raised in your review during the rebuttal process. With the discussion phase ending in a few hours, we kindly request any further feedback you may have on our responses.
>
> We look forward to your insights and appreciate your time and effort. Thank you.

---

### Official Review · Reviewer_hqBp · 2024-11-11

**Soundness:** 3
**Presentation:** 3
**Contribution:** 3
**Rating:** 6
**Confidence:** 3

**Summary:**

This paper studies pool-based active learning by developing a unified approach that performs well in both low-budget and high-budget regimes. While uncertainty-based approaches typically work better with higher budgets and representation-based methods with lower budgets, the paper proposes an algorithm that adaptively interpolates between the two. This is achieved through greedily selecting data points based on a notion called uncertainty coverage, which incorporates uncertainty into a representation-based objective. Empirically, the paper shows that the algorithm achieves or exceeds state-of-the-art performance across budget regimes.

**Strengths:**

- The problem of navigating active learning strategies across budget regimes seems interesting and relevant to the field, as highlighted by recent work [Hacohen and Weinshall, 2023].

- Balancing uncertainty and representativeness has been explored before in active learning, as discussed by the authors. The notion of uncertainty coverage introduced in this paper is built upon that of generalized coverage studied in [Bae et al., 2024] -- it is defined by weighting generalized coverage with the model's uncertainty. This idea is simple, intuitive, and seems versatile; the paper also briefly discusses connections to existing hybrid methods.

- The proposed approach seamlessly interpolates between representation-based and uncertainty-based approaches, without relying on a fixed budget threshold to switch behaviors (cf., [Hacohen and Weinshall, 2023]). The practical algorithm is also supported by some theoretical evidence such as approximation guarantees.

- The empirical validation of the algorithm demonstrates the interpolation, and the performance on various tasks/datasets seems promising across budget regimes.

**Weaknesses:**

- The theoretical results provide some insights but are primarily limited to a generalization guarantee for the empirical estimator of UCoverage and an approximation guarantee of the greedy algorithm. It would be interesting to see a deeper analysis that directly addresses the core goals in active learning, i.e., reducing error rates with as few labels as possible. For example, how does UCoverage perform as an objective in active learning in terms of *excess risk* and *label complexity* (potentially assuming oracle computation)?

- There does not seem to be an empirical comparison with SelectAL and TCM.

- The paper is generally well-written and easy to follow. However, there are a few ambiguities that may lead to confusion. For example, the term "budget" is never clearly defined, except in Algorithm 1, where $B$ is referred to as the budget. If this is the case, the discussion on parameter adaptation in Section 3.2 and parts of the experimental setup may be confusing, as the algorithm's behavior seems to depend on the current size of the labeled data rather than directly on $B$. Even if $B$ is small, $\mathcal{L}$ still grows over time and can be large later on. This raises the question of whether the budget refers to the total number of labeled data over $T$ steps, which was not clarified. I will also outline additional questions in the following section.

**Questions:**

- There is a discussion about discrete and continuous budget regimes in the comparison with SelectAL. In particular, it is mentioned that this paper cover continuous budget regimes. What do these regimes mean?
- There isn't a formal definition of the kernel function $k$ outside Proposition 4. Do you need to assume that $k$ is in a particular form and has a "spread" parameter, $\sigma$? Would the approach accommodate, e.g., polynomial kernel and sigmoid kernels? This is in reference to Definition 1, the discussion after Theorem 2, Algorithm 2, and the discussion before Proposition 4.
- In equation (1), it would be helpful to clarify what $N$ is.
- In Algorithm 1, what is ECE in line 2? Should $U(x_n)$ in line 5 be $U(x_n; f_{\tau^*})$?
- Can you elaborate on the computational complexity of Algorithm 1?
- The notation in Proposition 3 can be confusing -- what does $U(x;f) \rightarrow c$ exactly mean? Shouldn't $U(x;f)$ tend not to behave like constants as the model $f$ improves?
- Regarding Section 3.4, would you argue that the framework presented in this paper generalizes some existing methods?
- In Figure 3 (high-budget), why does the relative accuracy of UHerding drop as more labels are available?

---

> ### Author Response · Authors · 2024-11-21
> **Response, part 1**
>
> Thank you for your feedback and your work in reviewing our paper.
>
> ## Weaknesses
>
> > The theoretical results provide some insights but (...)
>
> We agree that, ideally, we would be able to provide theory about how UHerding reduces generalization error of an arbitrary model. In one special case, we can: because UCoverage is a generalization of Generalized Coverage (GCoverage), when using a constant uncertainty function and training a 1-nearest-neighbor classifier, we can exploit the generalization guarantee of Bae et al. (2024) to show generalization for the full learning process. Their Theorem 1, generalizing previous work, showed a generalization error bound as (1 - GCoverage) + GPurity; they also show that MaxHerding achieves an *estimated* GCoverage nearly as good as is possible to obtain. Our Corollary 6 accounts for the gap between the true and estimated GCoverage; plugging the two results immediately leads to a true generalization error bound for running MaxHerding with a 1-NN classifier in terms of the best obtainable GCoverage and the GPurity.
>
> The discussion at the end of Section 3.1 of Bae et al. (2024) gives a connection of their 1-NN bound to that of high-dimensional linear models, but requires very strong assumptions on the setting. Direct learning-theoretic analysis for non-constant U functions also seems difficult, although we agree that it would be very interesting.
>
> > There does not seem to be an empirical comparison with SelectAL and TCM.
>
> We were eager to include the comparison with SelectAL (*Hacohen et al., 2024)*. Unfortunately, as mentioned in Section 1 (and footnote 1), the authors of SelectAL have not released their code yet (though they indicated in private communication that they plan to do so). We attempted to reproduce their results, but were not successful.
>
> The main issue we ran into (and could not solve) was repeatedly switching between predicted budget regimes. Although SelectAL assumes that there is one transition from low-budget to high-budget, the model we implemented based on their description alternates budget regimes multiple times. For example, although we expect the predictions on budget regimes look like (low, low, …, low, high, high, …, high) where once the model predicts “high”, it never predicts low again, the reproduced model yields sequences like (low, low, high, low, low, high, high, low, high) with intermittent “low” predictions after “high” has been predicted. This inconsistency poses a problem because applying uncertainty-based methods in low-budget regimes can significantly degrade performance, and similarly, using representation-based methods in high-budget regimes can have adverse effects. Our implementation of SelectAL therefore gave much worse accuracies than they reported, and we did not think reporting those numbers would be fair.
>
> In fact, there is no guarantee that SelectAL prevents the model from predicting “low” again after it starts predicting “high”. Although we asked some follow-up questions to one of the authors back in September, we have not yet heard back. After seeing your review, we have contacted the author again and will aim to include a comparison to SelectAL once we receive a response.
>
> We would also like to clarify about TCM (Doucet et al., 2024). This method simply combines TypiClust (TC) and Margin (M), switching between them based on heuristics. The authors conducted extensive experiments to determine the ideal transition points, applying TypiClust for a fixed number of steps before switching to Margin based on the setting (e.g., in Tiny and Low settings, they perform three steps of TypiClust before switching; in Medium settings, they perform two steps, and so on). While their experiments provide valuable insights, we believe their heuristic approach will not generalize well to other settings without similar in-depth experimentation, which would not be possible without labels.
>
>
>
> > There are a few ambiguities that may lead to confusion. For example, the term "budget" is never clearly defined, except in Algorithm 1, where B is referred to as the budget.
>
>
> Thank you for highlighting that our use of the term “budget” was unclear; we indeed used it both to refer to the size of the labeled set $| \mathcal{L} |$ and to the number of data points to query at each step $t$, $B_t$. We conducted a thorough editing pass to clarify this distinction. In the revised text (highlighted in blue color), we now consistently define the size of the labeled set $| \mathcal{L}_t |$ as the “budget” and use “query budget” to refer to $B_t$ at each step $t$. Please let us know if any parts of the revised paper remain unclear.

---

> ### Author Response · Authors · 2024-11-21
> **Response, part 2**
>
> ## Questions
>
>
> > There is a discussion about discrete and continuous budget regimes in the comparison with SelectAL. In particular, it is mentioned that this paper cover continuous budget regimes. What do these regimes mean?
>
> First, we want to clarify that this use of "budget" refers to the total number of labeled data points. Here we used the terms *discrete* and *continuous* to describe the nature of budget regime distinctions. SelectAL assumes only two budget regimes – low and high – implying a single transition from low to high. However, we argue that budget regimes are not inherently discrete, and, in particular, not binary (i.e., limited to just low and high). Rather than assuming a single or discrete transition between budget regimes, it is more natural to view this transition as continuous; there is no sharp transition between low-budget and high-budget regimes. Rather, we think the utility of representation-based approaches mostly decreases relatively smoothly with budget, while that of uncertainty-based approaches does the opposite. We thus proposed a method smoothly interpolating between low-budget and high-budget behavior. While we still refer to low, mid, and high budget regimes to loosely indicate different budget levels, these are merely ways to split up our analysis, and not visible in any way to the UHerding method.
>
>
> > There isn't a formal definition of the kernel function  $k$ outside Proposition 4. Do you need to assume that $k$ is in a particular form and has a "spread" parameter, $\sigma$?
>
> Thanks for highlighting this. Every result in the paper works if $k\_\sigma(\\mathbf{x},\\mathbf{x}'; g)$ is of the form $\psi( (g(\mathbf{x}) - g(\mathbf{x}') ) / \sigma)$, for any feature mapping $g : \\mathcal{X} \\to \\mathbb{R}^d$, and $\psi : \\mathbb{R}^d \\to [0, 1]$ which is Lipschitz, has $\\psi(0) = 1$, and satisfies for any unit-norm direction $t \in \mathbb{R}^d$ that $\\lim\_{a \\to \\infty} \\psi(a t) = 0$.
>
> Theorem 1 as submitted states the exact conditions it needs, which are weaker than those above (in particular, it doesn't need a parameter $\\sigma$). Proposition 4 was missing its assumption on $\psi$; we have added it in the revised version.
>
> No results in our paper require $k$ to be a positive-definite kernel, nor do we need it to integrate to one (or even be integrable) as in kernel density estimation. For instance, the equivalence of MaxHerding to ProbCover shown by Bae et al. (2024) relied on using $k$ as a top-hat function, $k\_\sigma(\mathbf{x}, \mathbf{x}'; g) = \mathbb{1}(\\|g(\\mathbf{x}) - g(\\mathbf{x}')|| < \sigma)$, which is not positive-definite. (Neither is it Lipschitz, which is why our generalization theory does not apply and ProbCover estimation is so hard.) Although in practice the kernel we use is positive-definite, we can use non-kernel functions for UHerding without violating our theoretical results as long as they are in the form above. We added the definition in a footnote of the revised manuscript for clarity.
>
>
>
>
>
> > In equation (1), it would be helpful to clarify what $N$ is.
>
> In (1), we use Monte Carlo (MC) approximation to estimate the expectation over the data distribution. Thus, $N$ is the number of Monte Carlo samples. In particular, both MaxHerding and us use the empirical data distribution for this MC approximation where the samples are from the unlabeled dataset. To clarify, we added “with $\mathbf{x}\_n \in \mathcal{U}$” to (1) in the revision.
>
>
>
>
> > In Algorithm 1, what is ECE in line 2? Should $U(x\_n)$ in line 5 be $U(x\_n; f\_{\tau^*})$?
>
> ECE in Algorithm 1 stands for Expected Calibration Error (Naein et al., 2015) as described in Section 3.2. To be more clear and comprehensive, we have added a definition of ECE in a footnote of the revised manuscript.
>
> Regarding line 5, we thank the reviewer for pointing it out; it is indeed clearer with $U(x\_n; f\_{\tau^*})$. We have also updated the relevant text in Section 3.2 to clearly define $f\_{\tau^*}$. Please let us know if it is still unclear.

---

> ### Author Response · Authors · 2024-11-21
> **Response, part 3**
>
> > Can you elaborate on the computational complexity of Algorithm 1?
>
> There are three different major sub-parts in Algorithm 1: 1) computation of adaptive parameters $\tau^\*$ and $\sigma^\*$, 2) computation of proximity measure $k\_\sigma(\mathbf{x}, \mathbf{x}’; g)$, and 3) computation of uncertainty measure $U(\mathbf{x}; f)$.
>
> The time complexity of computing the proximity measure is equivalent to in MaxHerding, $\mathcal{O}( { | \mathcal{U} |}^2 \cdot d)$, where $d$ is the dimension of features from the feature extractor $g$. Computing uncertainty measures takes $\mathcal{O}(  | \mathcal{U} | / b \cdot M) ) $ time, where $M$ is the time complexity of one forward pass of a classifier $f$ on one data point, and $b$ is the size of a batch used in inference.
> In practice, $\mathcal{U}$ is large enough that the additional cost to computing uncertainty measures is orders of magnitude faster than computing the proximity, so the overhead over MaxHerding in this phase is negligible.
>
> Thus, the additional computational overhead of UHerding in Algorithm 1 compared to MaxHerding is the computation of adaptive parameters. In particular, the computation of $\tau^\*$ is more demanding, due to re-training of a classifier $f$ on the split of the labeled set $\mathcal{L}\_t$. Re-training of $f$ in low-budget regimes is quite fast in practice; even in high-budget regimes, re-training of $f$ requires about 60-70% of computation of training $f$.
>
> We would like to emphasize that the computational overhead of UHerding is *substantially* lower than that of SelectAL, which requires many re-training runs to select the budget regime and cross-validate. More importantly, we believe that the computational cost for adaptation is well-justified, given the significant performance gains UHerding offers over existing active learning methods across wide range of budget regimes.
>
>
>
>
> > The notation in Proposition 3 can be confusing -- what does $U(x;f) \rightarrow c$ exactly mean? Shouldn't $U(x;f)$ tend not to behave like constants as the model $f$ improves?
>
> Proposition 3 says that if the uncertainty measure of all data points in the unlabeled set $\mathcal{U}$ approaches to a non-negative constant $c$, the estimated UCoverage approaches to the estimated GCoverage up to a constant. The simplest way to achieve this is to simply select an uncertainty function which is a constant; then UCoverage directly reduces to GCoverage.
>
> In low-budget regimes where the classifier $f$ is generally not reliable, the uncertainty measure computed using $f$ is also not reliable. Hence, we would like the uncertainty measure to be nearly constant in low-budget regimes,  thus reducing essentially to the coverage measure. Rather than hardcoding a constant, however, and requiring logic to decide when to stop, we achieve $U(\\mathbf{x}; f) \\rightarrow c$ in low-budget regimes with temperature scaling, as described in Section 3.2. This adaptively applies a high temperature in low-budget regimes (thus, $U(\mathbf{x}; f) \rightarrow c$), and a low temperature in high-budget regimes.
>
> You are right that as the model $f$ improves, it is indeed preferable for $U(x;f)$ to be distinguishable and not behave like a constant. The $\\to$ in Proposition 3 is not an arrow that goes "with time" in Algorithm 1; it is the opposite, addressing what happens e.g. temperature scaling chooses a higher value.

---

> ### Author Response · Authors · 2024-11-21
> **Response, part 4**
>
> > Regarding Section 3.4, would you argue that the framework presented in this paper generalizes some existing methods?
>
> The proposed uncertainty coverage indeed generalizes some existing hybrid methods such as weighted $k$-means (Zhdanov et al., 2019)  and ALFA-Mix (Parvaneh et al., 2022), but up to some modifications specified in Section 3.4:
>
>
> 1. Replace $k$-means with greedy kernel $k$-medoids (or MaxHerding); see Appendix D to find a justification of this change
> 2. Apply a self-supervised feature extractor $g$ instead of a feature extractor from a classifier $f$
> 3. Replace their uncertainty measures with:
>     1. Weighted $k$-means: $U(\\mathbf{x}; f) = U'(\\mathbf{x}; f) \\cdot \\mathbb{1}[U'(\\tilde{\\mathbf{x}}; f) \\geq \\nu]$  where $\\nu \\ge 0$ satisfies $\\sum\_{n=1}^N \\mathbb{1}[U'(\\mathbf{x}\_n; f) \\ge \\nu] = M$, a pre-defined number, and $U'(\\mathbf{x}; f)$ denotes the original uncertainty measure used in weighted $k$-means
>     2. ALFA-Mix — $U(\\mathbf{x}; f) = \mathbb{1} [ \\exists\\, \text{ class } j \\text{ s.t. }\\; \\hat{y} (\\alpha\_j(\\mathbf{x}) \\, g(\\mathbf{x}) + (1-\\alpha\_j(\\mathbf{x})) \\, \\bar g\_j; f ) \\neq \\hat{y}(g(\\mathbf{x}); f) ]$; please refer to Proposition 8 for the definition of each term
>
> By making these adjustments, weighted $k$-means and ALFA-Mix can be transformed into the structure of UHerding, but without parameter adaptation. However, parameter adaptation is a crucial component for improving the generalization performance; we believe this is why weighted $k$-means and ALFA-Mix perform significantly worse than UHerding, despite their similar formulations.
>
>
>
>
> > In Figure 3 (high-budget), why does the relative accuracy of UHerding drop as more labels are available?
>
> As more labeled data points become available, performance eventually reaches a saturation point, converging to the accuracy of a model trained on the full dataset. For example, CIFAR10 has 50k training samples, so the performance of all active learning methods will eventually approach that of a model trained on all 50k samples as the budget increases. CIFAR10 is a relatively easy task, so performance saturates earlier – meaning that the accuracy of a model trained on 10k samples is not substantially different from one trained on the full 50k samples. As a result, models trained with Random selection will perform similarly to those trained with active learning methods in high-budget regimes, which explains the drop of relative accuracy (the improvement over Random) as the budget grows.

---

> ### Comment · Reviewer_hqBp · 2024-11-27
>
> Thank you for your detailed responses. I am happy to maintain my positive rating.

---

> > ### Author Response · Authors · 2024-11-28
> > **Response to Reviewer hqBp**
> >
> > We appreciate the reviewer for taking the time to check the rebuttal. The reviewer's constructive feedback has been invaluable in helping us improve our work. Thank you.

---

### Author Response · Authors · 2024-11-25
**Request for Feedback on Rebuttal**

Dear Reviewers,

We sincerely appreciate your thoughtful and constructive feedback on our work. As the discussion phase is ending soon in about two days, we kindly request any additional thoughts or comments on our rebuttals.

We have made every effort to address your concerns and suggestions comprehensively. However, if there are any points we may have missed or could elaborate further, please let us know. We are committed to incorporating constructive feedback to enhance our work.

We look forward to your input and thank you for your time and efforts.

---

### Meta-Review · Area_Chair_1pGy · 2024-12-11

**Metareview:**

This work presents a new metric termed uncertainty coverage that is flexible enough to smooth between low and high label budget objective functions. It then designs a greedy algorithm to optimize the metric. Reviewers agree that this paper develops an interesting approach for active learning, backed with provable guarantees. In addition, the effectiveness over prior works is verified by empirical study. Overall, this is a solid work.

**Additional Comments On Reviewer Discussion:**

Reviewers had questions about the why the new metric is more general, and how the hyper-parameters can be chosen. These questions were addressed during rebuttal. Authors also addressed questions about experiments. Since this work is more theory-oriented, and reviewers agree that it is technically sound, the AC believes it is above the bar without AC-reviewer discussion.

---

### Decision · Program_Chairs · 2025-01-22

Accept (Poster)